# Coherence–Diffusion Dynamics: A Continuous-Semantic Interpretation of Transformer Language Models

## Abstract

Despite strong empirical performance, the principles governing how large language models organize and stabilize semantic meaning during inference remain poorly understood. We focus on three recurrent empirical patterns that are not readily explained by existing approaches: (i) paradoxically higher attention concentration at incorrect token positions, (ii) a depth-dependent sign reversal in the relationship between predictive uncertainty and instability for incorrect tokens, and (iii) convergence of multiple signals derived from distinct empirical sources to a narrow depth range. We introduce the Coherence–Diffusion Dynamics (CDD) framework, which interprets Transformer inference as the evolution of a latent semantic trajectory under the joint influence of coherence-restoring drift and stochastic variability. Rather than positing new phenomena, CDD provides a unifying, phenomenological account that organizes these patterns into a regime-structured view of inference-time dynamics. Within this framework, we formulate four falsifiable structural constraints (C1–C4) on observable proxies of instability, diffusion, and coherence. Constraints C1–C3 formalize the observed empirical regularities as joint consistency requirements that frameworks lacking regime-level structure would not be expected to account for all three simultaneously within a single mechanism. C4 constitutes a derived, testable implication arising from the non-linear interaction between drift and diffusion. We evaluate these constraints on GPT-2 Large using inference-time measurements and examine cross-architecture consistency on Pythia-1.4B. On GPT-2 Large, all four constraints (C1–C4) are satisfied. On Pythia-1.4B, C1 and C2 replicate, while C3 and C4 are partially supported due to a broadened convergence window and entropy-induced sparsity in deeper layers. These findings provide a falsifiable, empirically grounded account of semantic trajectory dynamics and demonstrate that inference-time behavior exhibits regime-level structure that cannot be explained by independent or single-proxy analyses.

## 1 Introduction

The rapid development of large language models (LLMs) has transformed contemporary artificial intelligence, enabling systems to generate coherent text, explain complex relationships, and sustain extended dialogue (Brown et al., 2020; OpenAI et al., 2024). Despite these advances, the internal processes through which LLMs organize, stabilize, and occasionally destabilize semantic meaning during inference remain poorly understood, and modern language models continue to function as effective but opaque systems (Rudin, 2019; Vig & Belinkov, 2019).

Existing interpretability approaches provide valuable insights into localized mechanisms, but leave open a fundamental question: how semantic representations evolve as a structured process across depth, and why this evolution sometimes leads to abrupt failures such as hallucination. Attention visualizations cannot determine functional contribution (Clark et al., 2019); probing classifiers reveal correlations without causal structure (Alain & Bengio, 2018); and mechanistic interpretability methods such as causal tracing (Meng et al., 2023) and path patching (Wang et al., 2022) focus on specific circuits rather than the global evolution

of semantic state. As a result, the question of how the aggregate semantic trajectory stabilizes, destabilizes, and undergoes qualitative transitions during a forward pass remains largely unresolved.

This gap becomes concrete when examining inference-time behavior. We identify three recurring empirical patterns that are not readily explained within existing frameworks:

(i) **Paradoxical attention concentration.** Incorrect predictions are associated with higher attention concentration than correct ones, contrary to the expectation that focused attention should correlate with accuracy.

(ii) **Depth-dependent coupling reversal.** For incorrect token positions, the relationship between predictive uncertainty and instability change reverses sign across depth: uncertainty is stabilizing in early layers but destabilizing in later layers, while correct positions remain consistently stabilizing.

(iii) **Cross-signal depth convergence.** Multiple signals derived from distinct empirical sources (trajectory divergence onset, uncertainty–instability coupling reversal, and sensitivity to attention pruning) converge within a narrow depth range.

Taken together, these patterns suggest the presence of regime-level structure in semantic trajectory dynamics. Crucially, the challenge is not merely to explain each pattern in isolation, but to account for their co-occurrence within a single forward pass. Frameworks lacking regime-level structure would not be expected to account for all three simultaneously within a single mechanism.

Motivated by this observation, we introduce the *Coherence–Diffusion Dynamics (CDD)* framework, which interprets Transformer inference as the evolution of a latent semantic trajectory under the joint influence of coherence-restoring drift and stochastic variability. CDD is not a mechanistic or causal model; rather, it is a phenomenological framework that provides a structured account of how stabilization and variability interact across depth. Importantly, CDD does not posit new empirical phenomena. Instead, it organizes the observed patterns into a unified dynamical picture and imposes additional testable structure on their joint behavior. Within this framework, we formulate four *falsifiable structural constraints* (C1–C4) on observable proxies of instability, diffusion, and coherence. These constraints play distinct epistemic roles. Constraints C1–C3 formalize the observed empirical regularities as joint consistency requirements: their role is to test whether a single regime-structured framework can account for all three patterns simultaneously. Constraint C4 constitutes a derived, testable implication of the framework, arising from the non-linear interaction between drift and diffusion.

We evaluate these constraints using inference-time measurements on GPT-2 Large and examine cross-architecture consistency on Pythia-1.4B. On GPT-2 Large, all four constraints are satisfied, while on Pythia-1.4B the constraints are partially supported. These findings provide convergent evidence for a regime-structured interpretation of semantic trajectory dynamics.

This work makes three primary contributions:

- We show that three recurrent empirical patterns in Transformer inference jointly suggest the presence of regime-level structure, and that no existing single-proxy framework accounts for their co-occurrence.

- We introduce the CDD framework, which organizes these patterns into a unified dynamical interpretation and formalizes them as falsifiable structural constraints.

- We provide empirical evidence that these constraints hold across models, and demonstrate that the framework yields a non-trivial additional implication (C4) beyond explanatory unification.

## 2 Related Work

Research on meaning representation in artificial systems has progressed through several distinct paradigms, each capturing different aspects of the phenomena exhibited by modern language models. The goal of

this section is not simply to survey these paradigms, but to identify a common gap: none of the existing approaches directly addresses how the aggregate semantic state stabilizes, undergoes regime-level transitions, and occasionally destabilizes during a forward pass.

## 2.1 From Static Representations to Transformer Architectures

Early approaches treated meaning as discrete symbolic relations, offering explicit interpretability but struggling with graded similarity and contextual variation (Newell & Simon, 1976; Bobrow & Woods). Distributed representations such as Word2Vec (Mikolov et al., 2013) and GloVe (Pennington et al., 2014) established that semantic similarity can be expressed as geometric proximity, but assigned each token a single context-independent vector. Subsequent work on contextual embeddings revealed that representations are anisotropic, geometrically organized, and concentrated on low-intrinsic-dimensional manifolds (Ethayarajh, 2019; Arora et al., 2017; Kataiwa et al., 2025), suggesting that meaning is better characterized as a continuously evolving state. However, characterizing the process by which this state evolves, and the conditions under which it destabilizes, lies beyond the scope of static geometric analyses.

Sequential architectures such as LSTMs and GRUs (Hochreiter & Schmidhuber, 1997; Cho et al., 2014) introduced context-sensitive updating but were limited by vanishing gradients (Pascanu et al., 2013) and fixed-capacity hidden states. The Transformer (Vaswani et al., 2023) addressed these limitations through self-attention, enabling structured interactions among all tokens at each layer and producing a layered evolution of meaning interpretable as incremental movement within a continuous semantic space. Mechanistic interpretability studies further show that Transformers operate through linear residual streams combined with attention-mediated transformations (Nanda et al., 2023), revealing structural regularities that suggest an analogy with discrete approximations of underlying continuous flows. This residual stream perspective provides an important structural foundation, but does not by itself characterize the dynamical regimes through which semantic trajectories pass during inference.

## 2.2 Interpretability and Inference-Time Analysis

Attention-based analyses visualize token interactions but cannot reliably identify functional influence (Jain & Wallace, 2019), and probing classifiers expose correlations between representations and linguistic features without establishing causal mechanisms (Zhang et al., 2017).

Recent advances in mechanistic interpretability have begun to address dynamic aspects of computation. Causal tracing (Meng et al., 2023) localizes where factual associations are stored and retrieved across layers, while path patching (Wang et al., 2022) traces how information propagates between attention heads and MLP blocks. Automated circuit discovery (Conmy et al., 2023) extends these methods to identify task-relevant subgraphs systematically. These approaches characterize which components mediate specific computations and how information flows between them.

However, mechanistic interpretability operates at the level of individual circuits and localized causal pathways. It addresses where a specific computation occurs, but not how the aggregate semantic state evolves globally across depth under the joint influence of all components. In particular, it does not characterize regime-level transitions in which the qualitative character of semantic dynamics changes at a specific depth—nor does it predict the three empirical patterns identified in Section 1 that motivate the present work. Decoding and sampling studies characterize output variability (Holtzman et al., 2020), while loss landscape analyses illuminate optimization behavior (Sagun et al., 2018; Fort & Ganguli, 2019), yet none directly addresses the inference-time regime structure that determines when and why semantic trajectories destabilize.

## 2.3 Continuous-Time Dynamics of Transformers

A parallel line of work interprets Transformer layers as discretizations of a continuous-time dynamical system. Geshkovski et al. (2025) develop a framework in which tokens evolve as interacting particles governed by attention-mediated dynamics on the unit sphere, establishing that tokens cluster toward a small number of attracting configurations in the long-time limit, with a metastable phase preceding eventual collapse to a point

mass. Sander et al. (2022) show that Sinkhorn-normalized attention implements a discretized Wasserstein gradient flow, converging to heat diffusion in the infinite-depth limit. More broadly, the ResNet/Neural ODE literature (Chen et al., 2019; Weinan, 2017) establishes that residual networks can be viewed as Euler discretizations of ODEs, a perspective that motivates the analogy adopted in the CDD framework (Section 3 and Section 4.5).

The continuous-dynamics literature shares with CDD the foundational commitment that layer-wise updates are meaningfully interpreted as steps of an underlying continuous process. However, three differences determine why this literature does not resolve the gap identified above.

(i) **Asymptotic vs. inference-time.** Geshkovski et al. and related works analyze long-time or infinite-depth limits. CDD targets finite-depth inference dynamics in trained models: what happens during a typical forward pass, rather than the asymptotic regime that may require orders of magnitude more steps to reach.

(ii) **Idealized vs. real models.** The particle-system and gradient-flow analyses work with simplified Transformer variants and prove convergence results for these constructions. CDD operationalizes its parameters through directly measurable signals in real, trained models, without simplifying assumptions.

(iii) **Clustering vs. metastable regime.** The particle-system perspective predicts asymptotic collapse—a result at odds with the practical diversity of generated text. Geshkovski et al. attribute this tension to a long metastable phase. CDD models precisely this metastable regime, including the conditions under which coherence fails.

### 2.4 Eigenspectral Analysis of Representational Geometry

Jha & Reagen (2026) introduce NerVE, which tracks FFN eigenspectrum dynamics and shows that FFN nonlinearities actively redistribute variance across eigenmodes, with early layers exhibiting spectral bottlenecks and later layers showing broader latent-space utilization. This line of work operates at a different level of description from CDD: eigenspectral metrics characterize how variance is distributed across representation subspaces at a given layer, whereas CDD characterizes how the semantic state evolves across layers under the joint influence of coherence-restoring and variability-inducing dynamics. The two perspectives are complementary; whether the non-monotonic spectral patterns documented by NerVE correspond to the regime transitions identified in Section 5 is a direction for future investigation.

### 2.5 The Common Gap

Each research stream above has produced important insights, yet none explains why inference-time dynamics exhibit the specific empirical patterns observed, nor why signals derived from distinct empirical sources converge to a shared critical depth range. The three patterns identified in Section 1 make this gap explicit:

- Pattern I (paradoxical attention concentration for incorrect positions) cannot be explained by coherence- or circuit-level accounts, which implicitly assume a monotonic relationship between attention focus and prediction quality. Explaining the observed inversion requires distinguishing between regimes in which high attention concentration is stabilizing and those in which it leads to premature or misaligned commitment.

- Pattern II (the depth-dependent reversal in the relationship between predictive uncertainty and instability for incorrect positions) requires a regime-level account of how dynamics change across depth, which existing approaches do not provide. Attention analyses and probing classifiers do not distinguish early- from late-layer dynamics in this sense. Trajectory-tracking tools such as the logit lens and tuned lens (nostalgebraist, 2020; Belrose et al., 2025) characterize what changes across layers, but not why the relationship reverses at a specific depth. The continuous-dynamics literature, meanwhile, focuses on asymptotic behavior rather than the finite-depth regime transitions observed here.

- Pattern III (the convergence of three signals derived from distinct empirical sources to a common critical depth range) is not accounted for by any single-proxy analysis. It suggests the presence of an underlying structural transition, which smooth, monotone stabilization accounts do not anticipate.

CDD is designed to fill this gap. It does not replace existing approaches, but adds a missing layer of explanation: a unified dynamical framework that characterizes regime-level structure in semantic trajectory evolution and yields falsifiable constraints on observable behavior. In this sense, CDD complements existing methods by explaining not only what changes across depth, but why these changes take the structured form observed empirically.

## 3 The Coherence–Diffusion Dynamics Framework

Transformer hidden states reside in high-dimensional continuous vector spaces and evolve through repeated residual updates of the form $x_{t+1} = x_t + \Delta x_t$. This update structure admits a close analogy with Euler-type discretizations of continuous dynamical systems, in which a state advances through repeated small steps governed by an underlying vector field. Under this analogy, layer depth plays the role of a discretized time variable, and the sequence $x_0 \to x_1 \to \cdots \to x_L$ approximates a continuous semantic trajectory on an effective manifold $\mathcal{M} \subset \mathbb{R}^d$.

Empirical studies indicate that the intrinsic dimensionality of Transformer representations is substantially lower than the ambient dimension (Ethayarajh, 2019; Kataiwa et al., 2025), and that residual updates operate linearly within a shared vector space (Nanda et al., 2023). These observations motivate an interpretive perspective in which inference is not a sequence of isolated symbolic transformations, but a structured dynamical process shaped by coherence-restoring tendencies and stochastic variability. This dynamical lens helps explain why deeper layers exhibit progressive representational stabilization, why small perturbations can propagate into qualitatively different outcomes, and why models may undergo abrupt transitions into hallucination-like regimes (Holtzman et al., 2020).

The Coherence–Diffusion Dynamics (CDD) framework formalizes this perspective through three modeling assumptions (Section 3.1) and a continuous-time formulation (Section 3.2). The framework derives its scientific value not from the interpretive vocabulary alone, but from the falsifiable structural constraints it imposes on the joint behavior of observable proxies—constraints that organize the empirical patterns identified in Section 1 into a unified regime-structured account and impose additional testable structure on their joint behavior. These constraints are derived and stated explicitly in Section 3.3. Concrete connections to Transformer architecture are developed in Section 4.

### 3.1 Modeling Assumptions

The following three assumptions define the effective modeling regime within which the CDD interpretation is intended to apply. They are empirically motivated and abstract away architectural details while retaining the structural features shared across Transformer variants.

#### 3.1.1 Assumption A1: Continuous Semantic State

**Statement.** The semantic content of a sequence can be represented by a state variable $x$ taking values in a high-dimensional continuous space. Meaning is encoded in the geometric structure and relative position of representations within a semantic state space, rather than in discrete symbolic identities.

**Rationale.** Token embeddings and contextual states occupy structured, smoothly varying regions of $\mathbb{R}^d$, where semantic similarity is reflected in geometric proximity. Representations evolve smoothly across layers, forming trajectories well-approximated as continuous.

**Role.** A1 provides the representational foundation for CDD, allowing the latent semantic state to be modeled as a time-dependent variable $x(t)$ and enabling the introduction of an effective potential governing semantic stability, together with corresponding gradient-like forces that guide the evolution of the state.

### 3.1.2 Assumption A2: Context-Dependent Interaction

**Statement.** The semantic value of a representation is dynamically shaped by its interactions with other contextual representations. Semantic meaning is relational and evolves continuously as contextual information is integrated across the sequence.

**Rationale.** Self-attention mechanisms compute weighted interactions between representations, so the same token may assume substantially different semantic roles depending on its surrounding context. Semantic meaning thus emerges through ongoing relational modulation rather than from intrinsic properties of isolated representations.

**Role.** A2 motivates modeling semantic evolution as a context-driven, time-dependent process in which the trajectory $x(t)$ reflects the cumulative effect of relational interactions.

### 3.1.3 Assumption A3: Coherence-Driven Drift

**Statement.** Semantic evolution is subject to an effective coherence-restoring tendency that counteracts semantic disorganization. This tendency is modeled as a gradient-driven drift with respect to an effective potential $\Psi$ defined over the semantic state space:

$$\text{drift} = -\alpha \nabla \Psi(x).$$

Here, $\Psi$ is not assumed to be explicitly computable, but serves as an interpretive construct capturing the degree of semantic instability, and $\nabla\Psi$ represents the corresponding direction of increasing instability. The operator $\alpha$ represents the strength and structure of coherence enforcement and may act isotropically (scalar-valued), anisotropically (matrix-valued), or heterogeneously across layers or representational subspaces.

**Rationale.** Transformer models exhibit a strong preference for coherent and contextually appropriate continuations: perturbations to the semantic state often lead to counteracting effects that restore alignment with learned patterns of meaning. Modeling these as a gradient-driven flow toward regions of lower $\Psi$ provides a compact description of semantic stabilization.

**Role.** A3 defines the deterministic drift component of CDD. Together with A1 and A2, it enables semantic evolution to be modeled as an effective dynamical process shaped by contextual interactions and coherence-restoring forces.

### 3.1.4 Dynamic Sparsity as a Structural Consequence

Assumptions A1–A3 characterize semantic evolution as a continuous, relational, and coherence-driven process. Under this perspective, it is natural to expect that not all attention interactions contribute equally to the semantic trajectory. In particular, coherence-restoring dynamics (A3) suggest that semantic evolution may become increasingly concentrated on a restricted subset of high-impact interactions.

A formal stability property relevant to this intuition is given by Proposition 1. Importantly, this result is independent of the CDD assumptions A1–A3. It establishes that suppressing attention interactions whose total weight is $O(\varepsilon)$ induces only $O(\varepsilon^2)$ changes in the output distribution, under standard regularity conditions.

*Proposition* 1 (Stability under Low-Impact Attention Suppression). Let $S \subset \{1, \ldots, n\}$ be a set of attention interactions with total weight $\sum_{j \in S} a_{ij} \leq \varepsilon$. Under mild regularity conditions on the semantic update and output maps, suppressing such interactions induces only a second-order change in the output distribution:

$$D_{\text{KL}}(p \,\|\, p') = O(\varepsilon^2),$$

where $p$ and $p'$ denote the output distributions before and after suppression, respectively. A formal statement and proof are provided in Appendix A.1.3.

Proposition 1 provides a local stability guarantee: low-weight interactions can be removed with negligible impact on the output. This result does not imply that semantic influence is globally concentrated. However, within the CDD framework, it supports the following interpretation. As coherence-restoring effects stabilize the trajectory across depth, the effective contribution of attention interactions becomes increasingly concentrated, and a large fraction of low-weight interactions become dynamically irrelevant. This interpretation is not implied by Proposition 1, but is consistent with it. We therefore distinguish between local stability, as formalized by Proposition 1, and global, depth-dependent concentration of semantic influence, which we refer to as *dynamic sparsity*. The latter is examined empirically in Experiment III (Section 5.4).

### 3.2 Continuous-Time Formulation

#### 3.2.1 Instability Potential $\Psi$

The instability potential $\Psi$ is not directly observable. Token-level surprisal, loss spikes, and entropy irregularities serve as partial empirical indicators that correlate with elevated $\Psi$, without uniquely determining its value.

For the purposes of empirical analysis, we operationalize $\Psi$ using token-level surprisal:

$$\Psi_{\ell,t} := -\log p(y_t^* \mid h_{\ell,t}),$$

where $y_t^*$ denotes the ground-truth token at position $t$ and $p(\cdot \mid h_{\ell,t})$ is computed via a logit lens applied to the intermediate hidden state (see Section 5.2 for details). This choice provides a consistent readout of predictive uncertainty across layers without introducing additional learned parameters.

While $\Psi_{\ell,t}$ does not provide direct access to the gradient $\nabla\Psi$, its layer-wise trend serves as a practical indicator of whether semantic evolution is progressing toward or away from stable, low-$\Psi$ regions, aligned with the qualitative predictions of the CDD framework.

#### 3.2.2 Drift–Diffusion Equation

Transformer inference involves both stabilizing and variability-inducing effects. On one hand, representations tend to evolve toward semantically coherent configurations; on the other hand, uncertainty and competing continuations introduce variability into the evolution of hidden states. To capture this interplay at an abstract level, we adopt an effective drift–diffusion formulation:

$$dx(t) = -\alpha\nabla\Psi(x(t))\,dt + \sigma\,dW(t), \tag{1}$$

where $\Psi$ denotes the instability potential, $\alpha$ controls the strength of coherence-restoring dynamics, $\sigma$ captures stochastic variability, and $W(t)$ is a standard Wiener process. The coherence coefficient $\alpha$ may act as a scalar or as a matrix-valued operator reflecting anisotropic coherence enforcement.

This equation is not intended as a literal description of Transformer computation. Rather, it serves as an effective model that summarizes qualitative regularities observed in layer-wise representation dynamics. In particular, it provides a structured way to interpret how stabilization and variability interact during inference.

To connect this formulation to measurable quantities, we relate each term to empirically accessible proxies. The instability potential $\Psi$ is operationalized via token-level surprisal $\Psi_{\ell,t} = -\log p(y_t^* \mid h_{\ell,t})$ as defined above. The diffusion strength $\sigma$ is associated with predictive uncertainty, estimated using entropy-based measures of the output distribution. The coherence coefficient $\alpha$ reflects the degree to which layer-wise updates are aligned with reductions in predictive instability, and can be empirically related to attention concentration and the directional consistency of updates. We note that the instability proxy $\Psi_{\ell,t}$ and the diffusion proxy $\sigma_{\ell,t}^{(H)}$ are both derived from the logit-lens predictive distribution and are therefore structurally correlated; $\alpha_{\ell,t}^{\text{attn}}$ is derived from the attention weight distribution and constitutes a partially independent signal. Accordingly, $\Psi$, $\sigma$, and $\alpha$ should be understood as correlated empirical proxies for distinct qualitative aspects of the dynamics, rather than as independently measured dynamical variables.

While these quantities do not provide direct access to the continuous-time dynamics in equation 1, they enable a discrete, layer-wise interpretation of the drift–diffusion decomposition. In particular, changes in $\Psi_{\ell,t}$ across layers serve as an observable signal of semantic stabilization or destabilization, while entropy-based measures capture the relative strength of variability. Within this interpretation, the drift term corresponds to systematic reductions in $\Psi$, and the diffusion term corresponds to fluctuations induced by uncertainty in the predictive distribution.

Different relative magnitudes of the coherence and diffusion terms give rise to distinct qualitative regimes:

- **Drift-dominated regime**: $\Psi$ decreases consistently across layers, indicating progressive stabilization toward coherent semantic states.

- **Balanced regime**: $\Psi$ exhibits moderate fluctuations while remaining bounded, corresponding to controlled semantic exploration.

- **Diffusion-dominated regime**: $\Psi$ increases or fluctuates irregularly, reflecting unstable semantic evolution and increased risk of incoherent outputs.

- **Excessive coherence regime**: updates are dominated by strong drift forces, potentially leading to overshooting or oscillatory behavior in which $\Psi$ fails to decrease monotonically despite high $\alpha$. This regime predicts that attention concentration should be higher for incorrect than correct positions— a counterintuitive pattern that is empirically confirmed in Section 5.2 and constitutes falsifiable constraint C1 (Section 3.3).

### 3.2.3 Fokker–Planck Perspective

The collective behavior of semantic trajectory ensembles can be described by the Fokker–Planck equation corresponding to equation 1:

$$\frac{\partial \rho(x,t)}{\partial t} = \nabla \cdot \big(\alpha \rho(x,t)\nabla\Psi(x)\big) + \frac{\sigma^2}{2}\nabla^2\rho(x,t),$$

where $\rho(x,t)$ denotes the probability density over semantic states. When coherence-restoring drift dominates, probability mass concentrates near low-$\Psi$ regions, consistent with the progressive representational stabilization observed in deeper layers. This ensemble perspective clarifies why semantic stability emerges robustly despite local stochastic variability: the collective bias of $\rho(x,t)$ toward stable regions induces an effective stabilizing influence on individual trajectories.

### 3.2.4 Empirically Testable Implications

The drift–diffusion formulation yields the following qualitative predictions, each of which is stated here in general terms and made precise as a falsifiable constraint in Section 3.3:

1. **Stabilization toward coherent regions.** When coherence-restoring effects dominate, semantic trajectories concentrate in low-$\Psi$ regions associated with stable representations. (*Formalized as C3; examined in Experiments I–III.*)

2. **Regime shifts under increased diffusion.** As stochastic influence grows, trajectories drift away from coherent regions, corresponding to hallucination-like behavior. (*Formalized as C2; examined in Experiment II.*)

3. **Overshooting under excessive coherence enforcement.** Excessively strong coherence-restoring forces may produce oscillatory or unstable trajectories, manifesting as paradoxically elevated attention concentration for incorrect predictions. (*Formalized as C1; examined in Experiment I.*)

4. **Concentration of semantic influence.** As trajectories stabilize, the effective dimensionality of semantic evolution contracts, causing many representational components to become dynamically irrelevant. (*Formalized as part of C3; examined in Experiment III.*)

These qualitative implications motivate the empirical investigations in Section 5. Their precise falsifiable forms, together with the epistemic role of each constraint and the conditions under which each would be violated, are stated in Section 3.3.

### 3.3 Falsifiable Structural Constraints

The drift–diffusion formulation in Section 3.2 provides a qualitative description of semantic trajectory dynamics. The empirical patterns introduced in Section 1 (paradoxical attention concentration, depth-dependent coupling reversal, and cross-signal depth convergence) motivate the need for a unifying explanatory framework. Rather than treating these patterns as independent observations, we formalize them as a set of *falsifiable structural constraints* on the joint behavior of the observable proxies $(\Psi, \sigma, \alpha)$.

The role of the CDD framework is not to discover these empirical regularities in isolation, but to explain why they arise jointly and to impose additional testable structure on their co-occurrence. Accordingly, we distinguish two categories of constraint.

**Structured formulations (C1–C3).** These constraints provide formal restatements of the motivating observations within a regime-structured vocabulary. Their value lies in the explanatory unification they provide: we demonstrate that a single dynamical framework can account for all three patterns simultaneously, and that each constraint imposes non-trivial structure beyond its raw motivating observation.

**Derived implication (C4).** This constraint is not contained in the motivating observations. It arises from the non-linear interaction between coherence-restoring drift and stochastic variability, and constitutes a testable implication whose violation would provide evidence against the framework beyond what the motivating patterns alone could establish.

**Epistemic role of the constraints.** C1–C3 are evaluated as joint consistency requirements: the claim is not that CDD predicts these patterns, but that frameworks lacking regime-level structure would not be expected to account for all three simultaneously within a single mechanism. C4 occupies a distinct epistemic position: it is a testable implication not entailed by the motivating observations, and its empirical confirmation provides evidence for the regime taxonomy beyond explanatory unification.

**Remark on falsification strength.** The falsification conditions for C1–C3 are structurally constrained but not fully parameterized: the framework specifies the direction and sign of expected effects, but does not determine the precise depth of transitions or the magnitude of effect sizes. This reflects the phenomenological scope of the framework, which prioritizes structural constraints over parameter-level prediction. More precise quantitative predictions would require additional mechanistic assumptions beyond the present formulation.

**C1: Depth-Dependent Inversion of the Coherence–Accuracy Relationship.**
**Statement.** The relationship between attention-based coherence $\alpha^{\mathrm{attn}}$ and prediction accuracy is non-monotonic across depth:

- In early layers, higher $\alpha^{\mathrm{attn}}$ is associated with lower accuracy.

- In later layers, higher $\alpha^{\mathrm{attn}}$ is associated with higher accuracy.

**Interpretation.** C1 formalizes the empirically observed attention paradox within a depth-structured regime vocabulary, interpreting early- and late-layer dynamics as qualitatively distinct rather than as points on a single monotone curve. Its non-trivial content lies in the requirement that a single proxy exhibit opposing predictive relationships across depth. This bi-directional constraint rules out any account that treats attention concentration as a globally monotonic indicator of coherence quality. The inversion need not coincide exactly with the depth at which the full regime transition occurs; rather, it may emerge earlier as a precursor signal of excessive coherence, with the full transition reflected in the multi-signal convergence specified by C3. This ordering is consistent with the phenomenological scope of the CDD framework, which characterizes the existence and qualitative direction of regime-level effects without prescribing their precise depths.

**Falsification condition.** C1 is falsified if the $\alpha^{\mathrm{attn}}$–accuracy relationship is monotonic across depth, or if no systematic inversion is observed.

**C2: Regime-Dependent Sign Reversal of** $\mathrm{corr}(\sigma, \Delta\Psi)$**.**
**Statement.** For incorrect token positions, the correlation between predictive uncertainty and instability change exhibits a depth-dependent sign reversal:

- Early layers: $\mathrm{corr}(\sigma, \Delta\Psi) < 0$

- Late layers: $\mathrm{corr}(\sigma, \Delta\Psi) > 0$

For correct positions, this correlation remains predominantly non-positive across depth.

**Interpretation.** C2 provides a structured formulation of the observed coupling reversal, interpreting it as a transition between exploration-dominated and instability-dominated regimes. Its non-trivial content lies in the asymmetry requirement: the sign reversal must occur for incorrect positions but not for correct ones. This asymmetry cannot be explained by models that assume a uniform stabilization process across trajectory types.

**Falsification condition.** C2 is falsified if no sign reversal is observed for incorrect trajectories, or if the same reversal pattern occurs equally for correct trajectories.

**C3: Multi-Signal Convergence at a Critical Depth Range.**
**Statement.** Multiple signals derived from distinct empirical sources exhibit coordinated transitions within a common depth range:

- the onset of trajectory divergence, derived from the residual alignment gap;

- the sign reversal of $\mathrm{corr}(\sigma, \Delta\Psi)$, derived from entropy and surprisal under the logit-lens distribution;

- the peak sensitivity to attention pruning, derived from attention weight distributions under tail-mass suppression.

**Interpretation.** C3 formalizes the empirically observed convergence of signals as a joint consistency requirement on the framework. Its non-trivial content lies in the requirement that a *single regime-transition mechanism* account for all three signals simultaneously. A framework that explains each signal through independent mechanisms would not be expected to produce such convergence.

We note that the first two signals share a common logit-lens readout path, while the third arises from attention pruning and constitutes a substantively distinct measurement source. The evidential strength of C3 therefore rests on the convergence between the trajectory-based signals and the pruning-based sensitivity, while the agreement between the first two signals provides internal consistency rather than fully independent confirmation.

**Falsification condition.** C3 is falsified if these signals vary independently across depth, or if no common transition range is observed.

**C4: Non-Additive Interaction Between Coherence and Variability.**
**Statement.** The joint effect of $(\alpha, \sigma)$ on prediction accuracy cannot be explained as a simple combination of their marginal effects. In particular, positions with high $\alpha$ and low $\sigma$ exhibit disproportionately higher accuracy than expected from independent contributions, while positions with low $\alpha$ and high $\sigma$ exhibit disproportionately lower accuracy.

**Interpretation.** Unlike C1–C3, C4 is not required to explain the motivating observations and does not follow from analyzing $\alpha$ or $\sigma$ in isolation. It arises from the non-linear interaction between coherence-restoring drift and stochastic variability, which jointly determine the regime of semantic evolution. The non-trivial content

Table 1: Correspondence between CDD core quantities and Transformer components.

| CDD Quantity | Transformer Observable | Interpretation |
| --- | --- | --- |
| $\alpha$ | Attention sharpness, gradient alignment, residual update strength | Magnitude and effective direction of coherence-restoring dynamics |
| $\sigma$ | Softmax entropy, logit-level uncertainty | Magnitude of stochastic semantic exploration |
| $\Psi(x)$ | Surprisal (NLL), loss spikes, entropy irregularities | Degree of semantic instability or incoherence |
| $\varepsilon$-threshold | Very low attention weights / negligible gradients | Operational boundary of effective irrelevance in semantic dynamics |

of C4 lies in its directional asymmetry: the drift-dominated configuration (high $\alpha$, low $\sigma$) is expected to yield accuracy gains that exceed additive expectations, and conversely for the diffusion-dominated configuration (low $\alpha$, high $\sigma$). This directional constraint is not shared by additive or monotonic alternatives. We note, however, that C4 follows structurally from the opposing roles of $\alpha$ and $\sigma$ in the drift–diffusion formulation; it should therefore be understood as a derived implication of the framework's regime structure rather than an independent prediction.

**Falsification condition.** C4 is falsified if prediction accuracy is well explained by additive or independent effects of $\alpha$ and $\sigma$, or if the observed interaction lacks the predicted directional asymmetry.

In Section 5, we evaluate these constraints empirically on GPT-2 Large and examine their cross-architecture consistency on Pythia-1.4B.

# 4 Architectural Realization of Coherence–Diffusion Dynamics

The abstract constructs of the CDD framework admit systematic interpretations in terms of measurable internal signals within Transformer architectures, with dynamic sparsity arising as a structural consequence examined in Section 3.1.4. Table 1 summarizes the correspondence between CDD quantities and their architectural observables. These correspondences are not exact or mechanistic equivalences; they provide a principled way to relate abstract dynamical constructs to quantities that can be measured and analyzed empirically.

Throughout this section, the term *drift* refers exclusively to an effective modeling construct summarizing coherence-restoring tendencies at an abstract dynamical level, and should not be conflated with empirical layer-wise displacement measures.

## 4.1 Instability Potential $\Psi$ as a Loss-Based Semantic Signal

Operationally, $\Psi(x)$ is not computed directly by the model (Section 3.2.1). We treat token-level surprisal, layer-wise loss fluctuations, and entropy-based irregularities as partial observables that correlate with increases in $\Psi$. Low-$\Psi$ regions are associated with semantic states in which next-token probability mass is strongly concentrated around contextually appropriate continuations, gradients propagate smoothly across layers, and attention patterns remain relatively stable with depth. High-$\Psi$ regions are empirically associated with sudden increases in token-level surprisal, unstable shifts in attention focus, and rapidly varying gradient patterns — signals that often precede or accompany hallucination-like output.

## 4.2 Coherence Operator $\alpha$: Attention Concentration and Residual Gradient Alignment

The coherence operator $\alpha$ reflects the combined effect of attention concentration and the alignment of residual updates with directions that reduce predictive uncertainty. As noted in Section 3.2.2, $\alpha$ may act as a scalar, capturing the overall magnitude of coherence enforcement, or as a matrix-valued operator, reflecting anisotropic enforcement across representational subspaces. The architectural realizations described below

correspond to both aspects: attention concentration, as measured by entropy-based proxies, captures the scalar strength of coherence enforcement, while the attention–MLP decomposition reflects its directional and structural components.

### 4.2.1 Attention Concentration

Self-attention updates token representations as $h_i' = \sum_j a_{ij} V_j$, where $a_{ij} = \text{softmax}(q_i k_j^\top)$. When attention distributions are sharply concentrated, representational updates are dominated by a small subset of contextually relevant tokens, consistent with a higher effective value of $\alpha$. Diffuse attention can spread semantic influence across many tokens, weakening directional coherence and increasing susceptibility to drift.

However, excessively high concentration may also lead to overshooting or oscillatory behavior, as described by the excessive coherence regime in Section 3.2.2. In such cases, attention may become overly concentrated on a restricted subset of contextual signals, which can limit effective integration of broader context and is qualitatively associated with degraded prediction accuracy.

The operationalization of $\alpha$ as attention concentration is grounded in an established line of empirical work documenting that attention distributions in Transformer models are not uniformly informative, but systematically concentrated on a small subset of contextually relevant positions. Michel et al. (2019) show that a large fraction of attention heads can be removed at inference time without significant performance degradation, implying that semantic computation is dominated by a compact subset of high-impact interactions—an observation directly consistent with the CDD operationalization of $\alpha$ as a measure of how strongly coherence-restoring dynamics concentrate representational updates. Vig & Belinkov (2019) further demonstrate that attention entropy varies systematically across layers and heads, with lower entropy corresponding to more focused, content-dependent attention patterns, providing empirical grounding for treating attention concentration as a layer-wise diagnostic of coherence enforcement.

Taken together, these results support the view that attention concentration serves as a meaningful, albeit imperfect, proxy for coherence-restoring dynamics: it tracks the degree to which representational updates are dominated by a focused subset of contextual interactions, consistent with the interpretation of $\alpha$ as governing the strength of coherence enforcement.

Importantly, the relationship between attention concentration and model behavior is not monotone. The entropy collapse phenomenon identified by Zhai et al. (2023) demonstrates that excessively concentrated attention can destabilize model behavior—a finding originally reported in the context of training dynamics, but conceptually consistent with the inference-time excessive coherence regime described in Section 3.2.2, in which overly strong drift produces overshooting rather than stable convergence. This non-monotonicity is critical: it implies that attention concentration cannot be interpreted as a simple monotone indicator of improved coherence or accuracy.

Within the CDD framework, this non-monotonicity plays a structurally necessary role and gives rise directly to falsifiable constraint C1 (Section 3.3). The depth-dependent inversion in the relationship between $\alpha^{\text{attn}}$ and predictive accuracy follows naturally from the coexistence of drift-dominated and excessive-coherence regimes: $\alpha^{\text{attn}}$ should be associated with lower accuracy in early layers, where excessive coherence produces overshooting, and with higher accuracy in deeper layers, where elevated concentration reflects stable drift-dominated convergence. This is not consistent with a monotonic coherence–accuracy account, and constitutes a structurally non-trivial requirement of the CDD framework. The depth-dependent pattern is examined in detail in Section 5.2.

### 4.2.2 Residual Gradient Alignment

When the residual update $\Delta x_t$ is directionally aligned with changes that reduce predictive instability, i.e., movements toward lower-loss or lower-surprisal regions, semantic coherence tends to be reinforced. This alignment does not imply that residual updates explicitly perform gradient descent on $\Psi$; rather, it reflects an aggregate tendency for updates to bias semantic evolution toward lower-$\Psi$ regions when attention, normalization, and residual scaling are well calibrated.

The use of residual gradient alignment as a proxy for coherence-restoring dynamics is grounded in prior empirical and theoretical work on the directional properties of residual updates in Transformer models. Jastrzębski et al. (2018) argue that residual connections encourage networks to perform iterative inference, in the sense that each layer updates the hidden state in a direction of decreasing loss. Belrose et al. (2025) reproduce and extend this analysis in large autoregressive language models, showing empirically that the cosine similarity between the residual update $\Delta h_{\ell,t}$ and the negative loss gradient is negative in at least 95% of layer–position pairs across Pythia 6.9B—indicating a consistent, albeit weak, directional bias toward loss-reducing updates throughout the forward pass.

Within the CDD framework, this empirical regularity provides the interpretive grounding for $D_{\ell,t}^{\mathrm{align}}$ as a proxy for coherence-restoring dynamics (Section 5.1). If residual updates exhibit a systematic tendency to reduce predictive loss under normal inference conditions, then deviations from this tendency, as measured by the sign and magnitude of the alignment between $\Delta h_{\ell,t}$ and $-\nabla_{h_{\ell,t}} \Psi_{\ell,t}$, provide a diagnostic signal of whether a given position is being drawn toward or away from coherent semantic regions.

Crucially, this signal is not merely descriptive. The presence of a consistent directional bias allows the CDD framework to formulate testable constraints on trajectory evolution (C1–C4; Section 3.3). In particular, the distinction between drift-dominated and divergence regimes depends on whether this alignment remains stabilizing or becomes misaligned with the ground-truth direction. A positive $D^{\mathrm{align}}$ for incorrect positions indicates that the residual update is more strongly aligned with the direction of the model's own incorrect prediction than with the ground-truth token, consistent with the CDD interpretation of directional commitment to an incorrect semantic well. This behavior cannot be explained by a simple monotonic loss-minimization perspective, under which updates would consistently align with the ground-truth direction, and is empirically confirmed in Experiment I (Section 5.2).

It should be noted that the cosine similarities reported by Belrose et al. (2025) are small in absolute magnitude (typically below 0.05), reflecting the high dimensionality of the hidden state space rather than a weak effect. As shown in that work, these values substantially exceed what would be expected from random vectors of the same dimensionality. The $D^{\mathrm{align}}$ values reported in Experiment I (mean = 0.017, peak = 0.036) are of comparable magnitude, and their directional consistency across layers and positions provides the empirical basis for treating alignment as a structural component of inference-time dynamics.

### 4.2.3 Attention–MLP Functional Decomposition

The hidden-state update of a Transformer layer can be decomposed as $f(x) = f_{\mathrm{attn}}(x) + f_{\mathrm{MLP}}(x)$. Within the CDD framework, these components play distinct dynamical roles:

- **Self-attention** implements interaction-driven dynamics, mediating information flow across tokens and governing the directional component of coherence enforcement, and thus corresponds to the anisotropic, operator-valued aspect of $\alpha$.

- **MLP block** contributes a token-local stabilizing update that refines and stabilizes token-internal semantic structure through nonlinear expansion and contraction, and thus corresponds to the intrinsic stability structure of the drift field.

Residual connections combine these two effects, so that attention shapes the interaction geometry of the drift field while MLP shapes its intrinsic stability structure.

### 4.3 Diffusion $\sigma$ as Stochastic Variability

The diffusion parameter $\sigma$ characterizes the strength of variability-inducing effects in the drift–diffusion formulation introduced in Section 3.2.2. At a conceptual level, $\sigma$ governs the extent to which semantic evolution deviates from coherence-restoring dynamics, introducing fluctuations into the trajectory of the latent semantic state.

Since $\sigma$ is not directly observable, it is interpreted through empirically accessible measures of predictive uncertainty. In particular, $\sigma$ is associated with entropy-based statistics of the output distribution, which

quantify the dispersion of probability mass across competing continuations. Higher entropy corresponds to greater uncertainty and thus stronger effective diffusion. We note that because entropy and the instability proxy $\Psi_{\ell,t}$ are both derived from the output distribution, a substantial structural correlation between them is expected; our interpretation therefore focuses on the directional consistency of these signals with CDD predictions rather than treating them as independent measurements.

Within the layer-wise framework, entropy provides an observable signal of the variability of semantic evolution. When entropy is low, the model's predictions are sharply concentrated, and updates are effectively deterministic, corresponding to a low-$\sigma$ regime. As entropy increases, the model assigns comparable probability to multiple alternatives, leading to greater variability in semantic trajectories.

Importantly, diffusion is present even in the absence of explicit stochastic decoding mechanisms. Intrinsic sources of variability include competition among nearly equal logits, amplification of residual activations, and representational ambiguity among plausible continuations. These factors induce a non-zero baseline level of effective diffusion.

From the perspective of the instability potential $\Psi$, increased diffusion is associated with reduced consistency in layer-wise changes of $\Psi_{\ell,t}$, often manifesting as fluctuations or irregular trends across layers. In this sense, $\sigma$ modulates the stability of semantic evolution by influencing how reliably trajectories move toward or away from low-$\Psi$ regions. In terms of the qualitative regimes defined in Section 3.2.2, elevated $\sigma$ relative to $\alpha$ corresponds to the diffusion-dominated regime, while low $\sigma$ corresponds to regimes in which variability is limited, with the specific behavior depending on the magnitude of $\alpha$ (e.g., drift-dominated or excessive coherence regimes).

A complementary perspective on the interpretive validity of $\sigma_{\ell,t}^{(H)}$ as an empirical proxy emerges from a bin-level analysis of predictive accuracy. When token positions are stratified into tertiles by their layer-wise entropy value, the lowest-$\sigma$ tertile tends to exhibit higher correct-prediction rates than the highest-$\sigma$ tertile across most transformer layers. This relationship becomes more pronounced in intermediate-to-deep layers, where the gap between the lowest and highest tertiles widens, coinciding with the depth range at which correct and incorrect trajectories diverge most rapidly (Section 5.2). Because this stratification is defined by the $\sigma$ level prior to observing the correctness label, the pattern provides directional support for the view that $\sigma_{\ell,t}^{(H)}$ carries predictive information rather than merely describing outcomes post hoc. We emphasize that this constitutes an observational correlation rather than a causal claim, consistent with the phenomenological scope of the CDD framework.

## 4.4 Dynamic Sparsity and $\varepsilon$-Thresholds

Proposition 1 (Section 3.1.4) establishes that suppressing attention interactions with total weight at most $\varepsilon$ induces only $O(\varepsilon^2)$ changes in the output distribution. For a given attention head, many attention coefficients contribute only marginally to the semantic update; when interactions are ranked by contribution and pruned according to a cumulative tail-mass criterion, a substantial fraction can be removed without altering the output distribution.

Within the CDD framework, this phenomenon is interpreted as a consequence of drift–diffusion dynamics: as coherence-restoring forces guide semantic trajectories toward stable regions across layers, the effective dimensionality of semantic computation contracts, rendering peripheral interactions dynamically irrelevant. Dynamic sparsity thus emerges as a structural feature of semantic evolution rather than an ad hoc engineering artifact. The layer-wise behavior of this phenomenon, and its connection to the trajectory stabilization observed in Experiment I and II, is examined empirically in Experiment III (Section 5.4).

## 4.5 Layer-wise Evolution and CDD Predictions

The discretized analogue of the CDD dynamics at the layer level is

$$x_{t+1} = x_t - \alpha \nabla \Psi(x_t) + \sigma \xi_t,$$

Table 2: CDD–Transformer structural mapping and qualitative predictions.

| CDD Condition | Expected Behavior |
|---|---|
| High $\alpha^\dagger$ | Overshooting updates, oscillatory instability, brittle behavior |
| Low $\alpha^\dagger$ | Diffuse attention, weak gradient alignment, semantic drift |
| High $\sigma$ | Diffusion-dominated behavior; increased hallucination risk |
| Low $\sigma$ | Highly deterministic or repetitive generation |
| Influence $< \varepsilon$ | Removable interactions; sparsity without semantic loss |
| Low $\Psi$ | Bias toward coherent semantic regions |
| High $\Psi$ | Elevated risk of semantic collapse or hallucination |

$^\dagger$ These characterizations apply primarily to earlier transformer layers. In intermediate and deeper layers, elevated $\alpha^{\mathrm{attn}}$ is instead associated with stable drift-dominated convergence and higher predictive accuracy, indicating that the interpretive role of $\alpha^{\mathrm{attn}}$ undergoes a qualitative shift across depth. This non-monotone behavior is formalized as falsifiable constraint C1 (Section 3.3).

where $\xi_t$ represents aggregated stochastic influence. Table 2 summarizes the qualitative predictions of the CDD framework as a function of the relative magnitudes of $\alpha$ and $\sigma$. These predictions motivate the controlled experiments in Section 5.

# 5 Experiments: Verification of Falsifiable Constraints

This section evaluates the four falsifiable constraints C1–C4 derived in Section 3.3, together with the structural consequence of dynamic sparsity.

The evaluation is organized as follows. Section 5.1 describes the base experimental pipeline used to generate layer-wise trajectory measurements. Section 5.2 (Experiment I) operates directly on this pipeline to extract trajectory-level signals. Section 5.3 (Experiment II) re-analyzes these measurements to study coherence–diffusion coupling. Section 5.4 (Experiment III) introduces an independent intervention to probe dynamic sparsity through attention pruning. Section 5.5 synthesizes the results across experiments to evaluate C3 (multi-signal convergence) and C4 (joint $\alpha$–$\sigma$ interaction). Section 5.6 examines cross-architecture consistency on Pythia-1.4B.

The core of this evaluation lies in C3, requiring multi-signal convergence from independent empirical origins. By drawing trajectory divergence onset from Experiment I, $\sigma$–$\Delta\Psi$ sign reversal from Experiment II, and pruning sensitivity peak from Experiment III, we identify a consistent depth range across all three. This shared convergence constitutes collective evidence that transcends the scope of any individual experiment.

## 5.1 Base Experimental Pipeline

This section defines the base measurement pipeline used to generate layer-wise trajectory data. The data are obtained from GPT-2 Large under a teacher-forcing protocol, using 200 sequences drawn from the C4 English validation set. Token positions are classified as correct (model argmax prediction matches the ground-truth next token) or incorrect (model prediction diverges from ground truth). A matched sampling strategy is applied, retaining an equal number of correct and incorrect positions per sequence.

At each layer $\ell$ and token position $t$, the following proxies are computed uniformly across all experiments.

**Instability proxy** $\Psi_{\ell,t}^{\mathrm{GT}}$: token-level surprisal of the ground-truth token under the logit lens distribution, $\Psi_{\ell,t}^{\mathrm{GT}} = -\log p(y_t^* \mid h_{\ell,t})$, as defined in Section 3.2.1.

**Diffusion proxy** $\sigma_{\ell,t}^{(H)}$: Shannon entropy of the logit lens distribution at layer $\ell$ and position $t$, serving as a measure of predictive uncertainty.

**Attention concentration proxy** $\alpha_{\ell,t}^{\mathrm{attn}}$: normalized concentration of the attention distribution, computed as $1 - H(\mathbf{a}_{\ell,t})/\log n$, where $H(\mathbf{a}_{\ell,t})$ denotes the entropy of the mean attention weights averaged across heads

at layer $\ell$ and position $t$, and $n$ is the number of attended positions. The self-position entry is excluded prior to normalization to avoid the trivial self-focus induced by causal masking.

**Residual alignment gap $D_{\ell,t}^{\text{align}}$:**

$$D_{\ell,t}^{\text{align}} = \alpha_{\ell,t}^{\text{residual,Model}} - \alpha_{\ell,t}^{\text{residual,GT}},$$

where each alignment score is the cosine similarity between the residual update $\Delta h_{\ell,t} = h_{\ell,t} - h_{\ell-1,t}$ and the negative logit-lens gradient with respect to the corresponding token:

$$\alpha_{\ell,t}^{\text{residual}} = \cos\left(\Delta h_{\ell,t}, -\nabla_{h_{\ell,t}} \Psi_{\ell,t}\right).$$

A positive value of $D_{\ell,t}^{\text{align}}$ indicates that the residual update is more strongly aligned with the direction of the model's prediction than with the ground-truth token. Divergence onset is detected when $D_{\ell,t}^{\text{align}}$ exceeds a threshold of 0.05 and remains above this threshold for at least two consecutive layers, with no temporal smoothing applied.

All gradient computations use the logit lens readout path described in Section 3.2.1, applying the model's final layer normalization and output head directly to each intermediate hidden state. The readout path was validated against the model's actual output logits across all 200 samples, confirming that the logit lens path reproduces the final-layer predictions with relative $\ell_2$ error of 0.0.

## 5.2 Experiment I: Trajectory Analysis

**Role in the argument.** Experiment I produces the first of three signals required for C3: the layer at which incorrect-position trajectories exhibit detectable directional commitment to an incorrect semantic well, measured via the residual alignment gap $D^{\text{align}}$. In the course of establishing this signal, the experiment also documents two empirical phenomena, paradoxical attention concentration (C1) and persistent $\Psi^{\text{GT}}$ divergence, that constitute the observational content C3 is required to unify within a single regime-structured account.

**Setup.** This experiment operates on a trajectory dataset derived from the base measurement pipeline described in Section 5.1. After matched sampling, the dataset consists of 15,370 correct and 15,370 incorrect positions, yielding a total of 1,106,640 layer-wise records across 36 transformer blocks (layers 0–35).

### 5.2.1 Result 1: Divergence of $\Psi^{\text{GT}}$ Trajectories

The most salient finding concerns the layer-wise evolution of $\Psi_{\ell,t}^{\text{GT}}$, which measures the surprisal of the ground-truth token at each layer (Figure 1A).

For correct positions, $\Psi^{\text{GT}}$ decreases monotonically from layer 0 through layer 34, where it reaches its minimum. A slight uptick is observed at layer 35 due to logit lens convergence. This pattern is consistent with the CDD prediction of progressive stabilization: as inference proceeds through successive transformer blocks, semantic representations converge toward coherent regions of low instability associated with the correct continuation.

For incorrect positions, the trajectory evolves in the opposite direction. Beginning from a comparable starting value at layer 0 (mean = 9.99), $\Psi^{\text{GT}}$ increases steadily through successive layers, reaching a mean of 30.56 at layer 32. This monotonic divergence indicates that the hidden state trajectory is moving progressively further from the ground-truth direction throughout the forward pass, consistent with the CDD interpretation of dynamical trapping within an incorrect semantic well.

The separation between the two trajectories widens substantially with depth: the mean difference is +0.63 nats at layer 0, +8.97 at layer 16, and +29.00 at layer 32. This progressive separation reflects a depth-dependent dynamical process, as predicted by the CDD framework.

At layer 35, both trajectories exhibit sharp transitions consistent with the logit lens measurement converging to the model's actual prediction probabilities. For correct positions, $\Psi^{\text{GT}}$ reaches its minimum (0.64); for incorrect positions, it drops abruptly from 30.56 to 4.98.

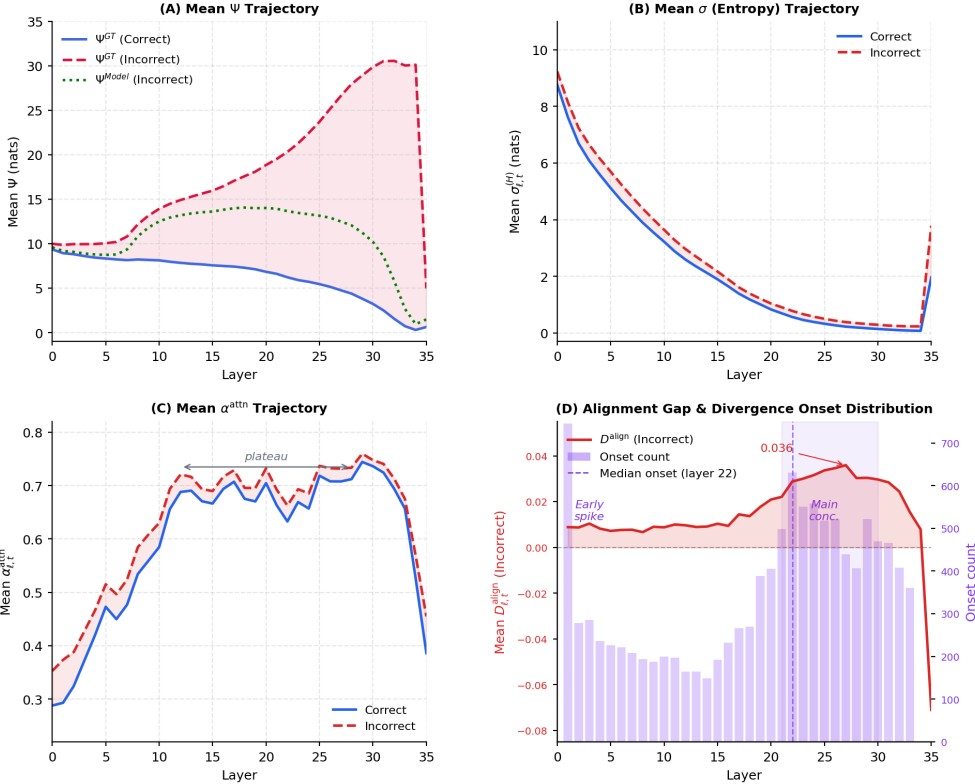

Figure 1: **Layer-wise trajectory of CDD proxies for GPT-2 Large (200 C4 sequences; 15,370 correct and 15,370 incorrect positions).** **(A)** Mean $\Psi_{\ell,t}^{\mathrm{GT}}$ trajectory. Correct positions decrease monotonically across layers, consistent with progressive stabilization; incorrect positions increase steadily, reflecting dynamical trapping within an incorrect semantic well. **(B)** Mean $\sigma_{\ell,t}^{(H)}$ (entropy) trajectory. Both conditions exhibit monotonically decreasing entropy; incorrect positions maintain consistently higher entropy at every layer. **(C)** Mean $\alpha_{\ell,t}^{\mathrm{attn}}$ trajectory. Both conditions show rapid concentration growth through layer 12 followed by a plateau. Incorrect positions exhibit slightly but consistently higher $\alpha^{\mathrm{attn}}$, consistent with the excessive coherence regime (Section 3.2.2). **(D)** Mean residual alignment gap $D_{\ell,t}^{\mathrm{align}}$ (red line, left axis) and divergence onset distribution (bars, right axis) for incorrect positions only. The gap remains positive throughout layers 1–34, indicating persistent directional commitment to an incorrect semantic well. The onset distribution is bimodal with a dominant concentration at layers 21–30 (median = 22; IQR: 11–27).

Across all incorrect positions, $\Psi^{\mathrm{GT}}$ exceeds $\Psi^{\mathrm{Model}}$ at every layer, with a mean gap of 7.72 nats across layers 1–34, confirming that the model assigns substantially lower surprisal to its own incorrect prediction than to the ground-truth token.

### 5.2.2 Result 2: Layer-wise Evolution of $\sigma$ and $\alpha^{\mathrm{attn}}$: The Attention Paradox

As Figure 1B shows, both correct and incorrect positions exhibit monotonically decreasing entropy across layers, consistent with the CDD prediction that stochastic variability contracts as semantic representations stabilize. Incorrect positions maintain consistently higher entropy than correct positions at every layer, indicating that representations associated with erroneous predictions retain greater diffuse variability throughout inference. We note that because $\sigma_{\ell,t}^{(H)}$ and $\Psi_{\ell,t}^{\mathrm{GT}}$ are both derived from the logit-lens distribution, a degree of structural correlation between them is expected; our interpretation therefore focuses on the directional consistency of these patterns with CDD predictions rather than treating them as independent measurements.

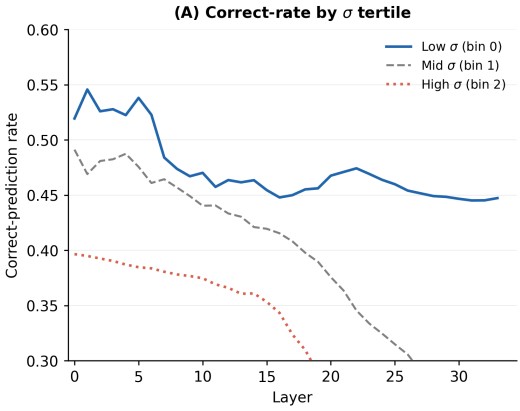
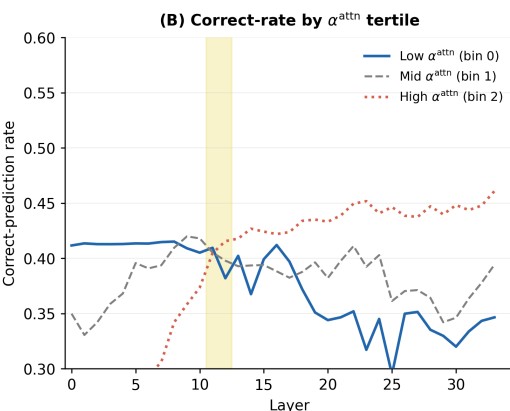

Figure 2: **Layer-wise correct-prediction rates stratified by tertile bins of (A) the diffusion proxy** $\sigma_{\ell,t}^{(H)}$ **and (B) the attention concentration proxy** $\alpha_{\ell,t}^{\mathrm{attn}}$. Bins are defined globally; bin 0 denotes the lowest tertile and bin 2 the highest. All results in this figure are computed over all token positions without matched sampling, and bin assignments are defined prior to conditioning on correctness. In (A), the lowest-$\sigma$ tertile exhibits consistently higher correct-prediction rates, with the gap widening from layer 12, consistent with C2. In (B), the highest-$\alpha$ tertile is associated with lower rates in early layers (0–10) and higher rates from layer 11 onward, constituting the empirical basis for C1.

A complementary analysis corroborates the interpretive validity of $\sigma_{\ell,t}^{(H)}$ as a proxy (Figure 2A): when token positions are stratified into tertiles by their $\sigma_{\ell,t}^{(H)}$ value, the lowest-$\sigma$ bin exhibits consistently higher correct-prediction rates than the highest-$\sigma$ bin across all layers. This monotone relationship holds from the earliest through the deepest transformer blocks, and the gap between bins widens in the intermediate-to-deep layers. Importantly, this analysis partitions positions by their $\sigma$ level *before* consulting the correctness label, providing directional support for the view that $\sigma_{\ell,t}^{(H)}$ anticipates predictive outcomes rather than merely describing them post hoc.

The primary evidence for C1 comes from the behavior of $\alpha^{\mathrm{attn}}$. Both conditions exhibit a rapid increase in attention concentration from layer 0 through approximately layer 12, followed by a plateau (Figure 1C). Contrary to the naive expectation that higher attention concentration would be associated with better predictions, incorrect positions exhibit slightly but consistently higher $\alpha^{\mathrm{attn}}$ than correct positions across all layers (differences ranging from 0.03 to 0.07). This pattern is consistent with the excessive coherence regime: overly strong drift forces produce premature commitment to incorrect semantic regions rather than stable convergence—the Transformer analog of premature convergence in physical systems under excessive drift (Section 6.1).

The depth-dependent inversion required by C1 is confirmed by the tertile analysis (Figure 2B): the highest-$\alpha$ bin exhibits *lower* correct-prediction rates than the lowest-$\alpha$ bin in early layers (approximately layers 0–10), consistent with the excessive coherence interpretation. This relationship *reverses* in intermediate and deeper layers (approximately layers 11 onward), where the highest-$\alpha$ bin is instead associated with higher correct-prediction rates, indicating that elevated attention concentration at these depths reflects stable drift-dominated convergence rather than overshooting.

### 5.2.3 Result 3: Residual Alignment Gap and Divergence Onset (C3 Signal 1)

The mean alignment gap $D_{\ell,t}^{\mathrm{align}}$ for incorrect positions is positive throughout most of the depth range, with a mean of 0.017 across layers 1–34 and layer-wise values reaching 0.036 at layer 27 (Figure 1D, red line). This indicates that the residual update $\Delta h_{\ell,t}$ is more strongly aligned with the direction of the model's incorrect prediction than with the ground-truth token, consistent with the CDD interpretation that the trajectory has been captured by an incorrect semantic well and is actively reinforcing that erroneous direction.

Table 3: Empirical outcomes from Experiment I. Rows are grouped by their role in the argument: C3 Signal 1 (primary contribution), C1 (phenomenon unified by C3), and supporting evidence for C2 and C4.

| Role | Structural Requirement | Observed Pattern | Assessment |
|---|---|---|---|
| *C3 Signal 1 (primary contribution of Experiment I)* | | | |
| C3 | $\Psi^{\mathrm{GT}}$ decreases monotonically (correct) | $9.37 \to 0.64$ (monotonic) | Consistent |
| C3 | $\Psi^{\mathrm{GT}}$ increases with depth (incorrect) | $9.99 \to 30.56$ (monotonic) | Consistent |
| C3 | Trajectory divergence onset concentrated at critical depth | Median onset layer 22; bimodal distribution | Consistent |
| *C1: attention paradox (phenomenon unified by C3)* | | | |
| C1 | $\alpha^{\mathrm{attn}}$–accuracy relationship inverts with depth | Highest-$\alpha$ tertile: lower accuracy in layers 0–10, higher from layer 11 | Consistent |
| *Supporting evidence (C2, C4)* | | | |
| C2, C4 | $\sigma$ decreases with depth; higher for incorrect positions | Monotonic decrease in both conditions; consistently higher at all layers | Consistent |

Of the 15,370 incorrect positions, 11,559 (75.2%) exhibit a detectable onset under the threshold criterion. Among positions with a detectable onset, the distribution is notably bimodal (Figure 1D, bars): a pronounced spike appears at layer 1 ($n = 745$; 6.4% of onset cases), and a second larger concentration spans layers 21–30 (44.3% combined), with a peak in layers 21–25 (23.9%). The median onset layer is 22 and the mean is 19.1 (SD = 9.78).

### 5.2.4 Summary

The primary contribution of Experiment I is *C3 Signal 1*: the divergence onset at layer 22 derived from residual alignment gap measurements (Table 3). The experiment also documents the attention paradox (C1), which emerges earlier in depth (around layer 11) and functions as a precursor signal, while the onset captures the structural commitment underlying C3.

### 5.3 Experiment II: Coherence–Diffusion Coupling Analysis

**Role in the argument.** Experiment II produces the second of three signals required for C3: the layer at which the coupling between predictive uncertainty and instability change reverses sign for incorrect positions, measured via $\mathrm{corr}(\sigma_{\ell,t}^{(H)}, \Delta\Psi_{\ell,t}^{\mathrm{GT}})$. In the course of establishing this signal, the experiment also documents the regime-dependent asymmetry between correct and incorrect trajectories (C2).

**Setup.** This experiment conducts a post-hoc coupling analysis on the token-level records produced by Experiment I, using the same matched position sets (15,370 correct and 15,370 incorrect positions). No additional model evaluation is required. The central derived quantity is the layer-wise instability change:

$$\Delta\Psi_{\ell,t}^{\mathrm{GT}} = \Psi_{\ell+1,t}^{\mathrm{GT}} - \Psi_{\ell,t}^{\mathrm{GT}},$$

where $\Delta\Psi_{\ell,t}^{\mathrm{GT}} < 0$ indicates a stabilizing update and $\Delta\Psi_{\ell,t}^{\mathrm{GT}} > 0$ indicates a destabilizing update. The stabilizing fraction at layer $\ell$ is defined as the proportion of positions for which $\Delta\Psi_{\ell,t}^{\mathrm{GT}} < 0$. Pearson correlations between each proxy and $\Delta\Psi_{\ell,t}^{\mathrm{GT}}$ are computed independently at each layer for correct and incorrect positions separately.

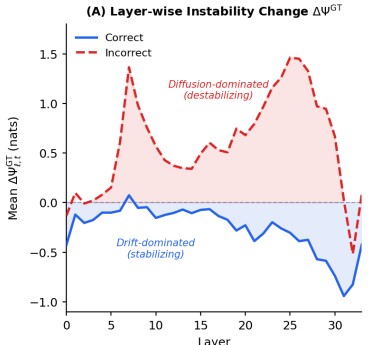 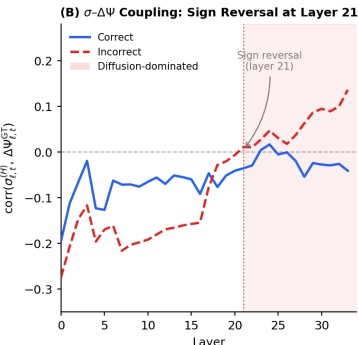 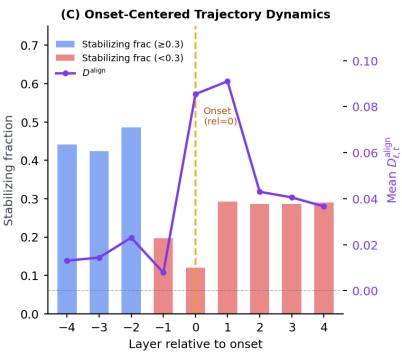

Figure 3: **Coherence–diffusion coupling analysis (GPT-2 Large, 15,370 correct and 15,370 incorrect positions). (A)** Layer-wise mean instability change $\Delta\Psi_{\ell,t}^{\mathrm{GT}}$ for correct (blue) and incorrect (red) positions. Correct positions exhibit $\Delta\Psi^{\mathrm{GT}} < 0$ at nearly every layer, confirming drift-dominated stabilization; incorrect positions exhibit predominantly positive $\Delta\Psi^{\mathrm{GT}}$, consistent with diffusion-dominated destabilization. **(B)** Layer-wise Pearson correlation between $\sigma_{\ell,t}^{(H)}$ and $\Delta\Psi_{\ell,t}^{\mathrm{GT}}$. For correct positions, the correlation remains predominantly negative throughout. For incorrect positions, the correlation reverses sign at layer 21 (shaded region), marking a transition to a diffusion-dominated regime—closely coinciding with the median divergence onset layer of 22 from Experiment I. **(C)** Onset-centered dynamics for incorrect positions with a detectable divergence onset ($n = 11{,}559$). The stabilizing fraction drops sharply at the onset while $D_{\ell,t}^{\mathrm{align}}$ spikes, indicating abrupt directional commitment to an incorrect semantic well rather than a diffusion-driven perturbation. Panels (A) and (B) display layers 0–33; layer 34 is excluded due to a logit-lens boundary artifact.

### 5.3.1 Result 1: Drift-Dominated Regime in Correct Positions

For correct positions, $\Delta\Psi_{\ell,t}^{\mathrm{GT}}$ is negative at nearly every layer from 0 to 33, with a single exception at layer 7 (mean = +0.07 nats), confirming that correct-position trajectories exhibit systematic stabilization throughout the forward pass (Figure 3A). The stabilizing fraction remains above 0.55 across layers 0–29, consistent with a drift-dominated regime in which coherence-restoring dynamics consistently outweigh diffusion-induced variability.

The correlation between $\sigma_{\ell,t}^{(H)}$ and $\Delta\Psi_{\ell,t}^{\mathrm{GT}}$ is negative across the full depth range, with the exception of a narrow deviation at layers 23–24 (Figure 3B). This indicates that, for correct positions, higher predictive uncertainty is consistently associated with stronger subsequent stabilization. In contrast, no consistent directional relationship is observed between $\alpha_{\ell,t}^{\mathrm{attn}}$ and $\Delta\Psi_{\ell,t}^{\mathrm{GT}}$ across layers, suggesting that attention concentration alone is not a reliable predictor of the direction of instability change within the drift-dominated regime.

### 5.3.2 Result 2: The Regime Transition Asymmetry (C3 Signal 2)

For incorrect positions, $\Delta\Psi_{\ell,t}^{\mathrm{GT}}$ is positive across nearly all layers, with the mean instability change increasing steadily to a peak of +1.46 nats at layer 25 before partially recovering in layers 28–33.

The central finding of this experiment concerns the layer-wise sign of $\mathrm{corr}(\sigma_{\ell,t}^{(H)}, \Delta\Psi_{\ell,t}^{\mathrm{GT}})$ and its asymmetry between correct and incorrect positions. For incorrect positions, this correlation is negative across layers 0–20 (range: $-0.27$ to $-0.007$), similar in direction to the correct-position pattern. However, beginning at layer 21, the correlation becomes positive and remains so through layer 33 (range: $+0.01$ to $+0.14$). Crucially, this reversal occurs only for incorrect positions; correct positions remain predominantly negative across depth, with only narrow localized deviations that do not amount to a sign reversal. This asymmetry (drift-dominated absorption for correct trajectories, diffusion-dominated amplification for incorrect ones) is the structural content of C2, and is not consistent with any uniform stabilization account.

Within the argument for C3, this sign reversal at layer 21 constitutes C3 Signal 2. Its near-coincidence with the median onset layer of 22 from Experiment I is examined in Section 5.5.

### 5.3.3 Result 3: Onset-Centered Dynamics

To further characterize the mechanism underlying the regime transition, we examine $\Delta\Psi_{\ell,t}^{\mathrm{GT}}$ and $\sigma_{\ell,t}^{(H)}$ in a window of $\pm 4$ layers centered on the detected onset layer for each incorrect position (11,559 positions with a detectable onset; Figure 3C).

Table 4: Onset-centered dynamics for incorrect positions with a detectable divergence onset ($n = 11{,}559$). Relative layer 0 corresponds to the onset layer.

| Rel. layer | $n$ | $\Delta\Psi^{GT}$ | Stab. frac. | $\sigma$ | $D^{\mathrm{align}}$ |
|---:|---:|---:|---:|---:|---:|
| $-4$ | 10,251 | $+0.34$ | 0.442 | 2.45 | 0.013 |
| $-3$ | 10,536 | $+0.40$ | 0.424 | 2.45 | 0.014 |
| $-2$ | 10,814 | $+0.15$ | 0.487 | 2.44 | 0.023 |
| $-1$ | 11,559 | $+1.15$ | 0.197 | 2.75 | 0.008 |
| **0** | **11,559** | **+1.74** | **0.121** | **2.50** | **0.085** |
| $+1$ | 11,558 | $+1.22$ | 0.265 | 2.19 | 0.091 |
| $+2$ | 11,198 | $+1.31$ | 0.261 | 2.00 | 0.046 |
| $+3$ | 10,790 | $+1.35$ | 0.256 | 1.84 | 0.045 |
| $+4$ | 10,324 | $+1.46$ | 0.260 | 1.71 | 0.041 |

Table 4 reports the mean values of each quantity at relative layers $-4$ through $+4$. Three patterns are noteworthy.

First, $\Delta\Psi_{\ell,t}^{\mathrm{GT}}$ increases sharply in the layer immediately preceding the onset (rel. layer $= -1$: mean $= +1.15$ nats) and reaches its maximum at the onset layer itself (rel. layer $= 0$: mean $= +1.74$ nats), before partially recovering in subsequent layers, confirming that the onset criterion captures a genuine local maximum of instability growth rather than an arbitrary threshold crossing.

Second, the stabilizing fraction drops sharply from 0.442 at rel. layer $= -4$ to 0.121 at rel. layer $= 0$, indicating a collapse of stabilizing behavior at the onset layer. Fewer than 13% of positions exhibit a stabilizing update at this depth, compared to a mid-depth baseline of approximately 44%.

Third, the most diagnostic evidence for the mechanism underlying C3 Signal 2 lies in the contrasting behavior of the two signals: $\sigma_{\ell,t}^{(H)}$ does *not* exhibit a sharp change around the onset (range: 1.71–2.75 across the window), whereas $D^{\mathrm{align}}$ spikes to 0.085 at the onset layer—approximately four times the baseline at rel. layer $= -4$ (0.013). This dissociation confirms that the regime transition is driven by abrupt *directional commitment* to the incorrect prediction rather than by a sudden increase in diffusion. The sign reversal of $\mathrm{corr}(\sigma, \Delta\Psi)$ is therefore a consequence of structural trajectory capture rather than a stochastic fluctuation, consistent with the CDD account of regime transitions as structural events. The physical analog is the irreversible capture of a trajectory in a local minimum: once commitment occurs, subsequent dynamics reinforce rather than correct the error (Section 6.1).

### 5.3.4 Summary

The primary contribution of Experiment II is *C3 Signal 2*: the sign reversal of $\mathrm{corr}(\sigma^{(H)}, \Delta\Psi^{\mathrm{GT}})$ for incorrect positions at layer 21. A similar reversal is observed over all positions at layer 19, indicating that the transition is not an artifact of incorrect-only conditioning. The experiment also establishes the regime asymmetry between correct and incorrect trajectories (C2). See Table 5 for detailed results.

### 5.4 Experiment III: Dynamic Sparsity Analysis

**Role in the argument.** Experiment III produces the third and final signal required for C3: the depth at which pruning sensitivity peaks and thereafter decreases broadly, measured via layer-wise KL divergence

Table 5: Empirical outcomes from Experiment II. Rows are grouped by their role in the argument: C3 Signal 2 (primary contribution), C2 (phenomenon unified by C3), and supporting evidence.

| Role | Structural Requirement | Observed Pattern | Assessment |
|------|------------------------|------------------|------------|
| *C3 Signal 2 (primary contribution of Experiment II)* | | | |
| C3 | Correct positions exhibit drift-dominated stabilization throughout | $\Delta\Psi^{\mathrm{GT}} < 0$ at all layers except layer 7; stab. frac. $> 0.55$ | Consistent |
| C3 | Sign reversal of $\mathrm{corr}(\sigma, \Delta\Psi)$ for incorrect positions coincides with trajectory divergence onset | Reversal at layer 21; onset median at layer 22 | Consistent |
| C3 | Not incorrect-only artifact | All-position reversal at layer 19 | Consistent |
| *C2: regime asymmetry (phenomenon unified by C3)* | | | |
| C2 | $\mathrm{corr}(\sigma, \Delta\Psi)$ reverses sign for incorrect positions; correct positions remain negative throughout | Sign reversal at layer 21 (incorrect only); correct positions remain predominantly negative throughout (32/34 layers) | Consistent |
| C2 | Onset driven by directional commitment, not diffusion spike | $D^{\mathrm{align}}$ spikes to 0.085; $\sigma$ stable at onset | Consistent |

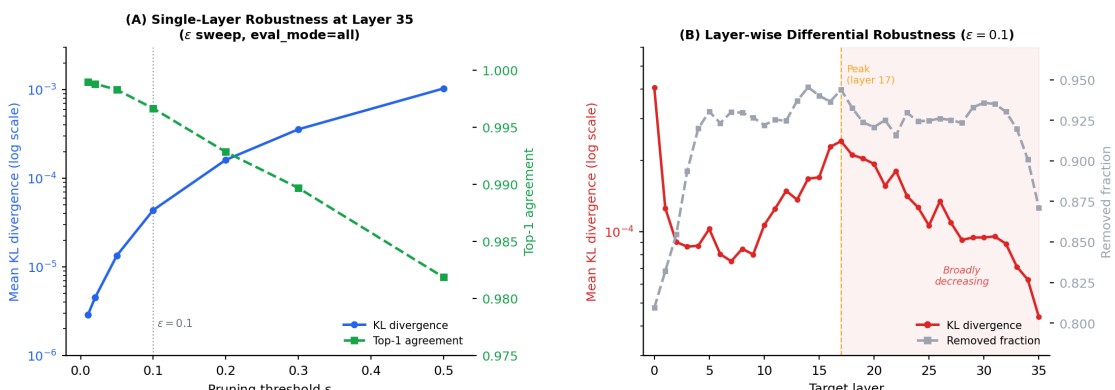

Figure 4: **Dynamic sparsity analysis under tail-mass pruning (GPT-2 Large, 200 C4 sequences, `eval_mode=all`, `keep_min=1`). (A)** Single-layer robustness at the final transformer block (layer 35) across pruning thresholds $\varepsilon \in \{0.01, 0.02, 0.05, 0.1, 0.2, 0.3, 0.5\}$. KL divergence remains below $10^{-3}$ even at $\varepsilon = 0.5$, where 48% of attention mass and 96% of attention entries are removed, while Top-1 agreement stays above 0.98 throughout, directly supporting Proposition 1. **(B)** Layer-wise KL divergence at fixed $\varepsilon = 0.1$. Layer 0 is substantially more sensitive than all other layers (KL $= 4.1 \times 10^{-4}$, approximately $9.3\times$ higher than layer 35). Among layers 1–35, KL reaches a local peak at layer 17 ($2.4 \times 10^{-4}$, approximately $5.5\times$ higher than layer 35), following the $\alpha^{\mathrm{attn}}$ plateau onset at layer 12 and preceding the regime transition at layer 21 identified in Experiment II. KL then decreases broadly from layer 17 to layer 35 (shaded region), consistent with trajectory stabilization. The layer-wise variation in robustness is not predicted by the Lipschitz bound of Proposition 1 alone, and constitutes C3 Signal 3.

under tail-mass suppression. This signal is derived from attention weight distributions, a substantively distinct measurement source from the residual alignment gap of Experiment I and the logit-lens entropy and surprisal of Experiment II, and therefore constitutes the strongest independent confirmation of the critical depth range. In the course of establishing this signal, the experiment also provides empirical support for

the structural consequence of *dynamic sparsity*: the progressive concentration of effective semantic influence toward a compact active core as trajectories stabilize across depth.

**Setup.**  We evaluate GPT-2 Large on 200 prompts from the C4 English validation set under teacher forcing, using float16 precision and a batch size of 8. Pruning is implemented as tail-mass suppression: for a given threshold $\varepsilon$, the smallest attention entries whose cumulative probability mass does not exceed $\varepsilon$ are set to zero, after which the remaining weights are renormalized. A minimum-keep constraint of `keep_min=1` is applied, ensuring that at least one attention entry is retained per query position after pruning, to prevent complete attention collapse. KL divergence and Top-1 agreement are evaluated across all query positions (`eval_mode=all`).

The experiment consists of two parts. Part A applies pruning exclusively at the final transformer block (layer 35), sweeping $\varepsilon \in \{0, 0.01, 0.02, 0.05, 0.1, 0.2, 0.3, 0.5\}$. This constitutes a direct empirical test of Proposition 1. Part B applies pruning independently at each layers with fixed $\varepsilon = 0.1$. This constitutes a stress test that substantially exceeds the scope of Proposition 1: it reveals layer-wise variation in robustness that the uniform $O(\varepsilon^2)$ Lipschitz bound does not predict, and produces C3 Signal 3.

### 5.4.1   Part A: Single-Layer Robustness (Support for Proposition 1)

Table 6: Part A: Threshold-invariant robustness at the final transformer layer (layer 35) under tail-mass pruning across all query positions.

| $\varepsilon$ | KL (mean) | Top-1 | Removed mass | Removed frac |
|---|---|---|---|---|
| 0.00 | 0.000 | 1.000 | 0.000 | 0.000 |
| 0.01 | $2.9 \times 10^{-6}$ | 0.999 | 0.009 | 0.766 |
| 0.02 | $4.5 \times 10^{-6}$ | 0.999 | 0.019 | 0.793 |
| 0.05 | $1.3 \times 10^{-5}$ | 0.998 | 0.048 | 0.835 |
| 0.10 | $4.4 \times 10^{-5}$ | 0.997 | 0.097 | 0.871 |
| 0.20 | $1.6 \times 10^{-4}$ | 0.993 | 0.193 | 0.910 |
| 0.30 | $3.6 \times 10^{-4}$ | 0.990 | 0.290 | 0.934 |
| 0.50 | $1.0 \times 10^{-3}$ | 0.982 | 0.480 | 0.963 |

Table 6 reports the results at the final transformer layer. Across all tested thresholds, the output distribution remains highly stable. At $\varepsilon = 0.1$, approximately 87% of attention entries and 9.7% of attention mass are removed, yet KL divergence is $4.4 \times 10^{-5}$ and Top-1 agreement is 0.997. Even at $\varepsilon = 0.5$, where 96% of entries and 48% of mass are suppressed, Top-1 agreement remains at 0.982. These results confirm that semantic computation at the final decoding step is governed by a compact active core, with the remaining interactions contributing negligibly to the output distribution, consistent with Proposition 1 (Figure 4A).

### 5.4.2   Part B: Layer-wise Differential Robustness (C3 Signal 3)

Table 7: Part B: Layer-wise robustness under tail-mass pruning with $\varepsilon = 0.1$. Layer 0 corresponds to the first transformer block; layer 35 to the last.

| Layer | KL (mean) | Top-1 | Removed mass | Removed frac |
|---|---|---|---|---|
| 0 | $4.1 \times 10^{-4}$ | 0.991 | 0.096 | 0.809 |
| 8 | $8.5 \times 10^{-5}$ | 0.995 | 0.093 | 0.930 |
| **17** | $\mathbf{2.4 \times 10^{-4}}$ | **0.991** | **0.091** | **0.934** |
| 24 | $1.3 \times 10^{-4}$ | 0.993 | 0.093 | 0.924 |
| 32 | $8.9 \times 10^{-5}$ | 0.995 | 0.092 | 0.930 |
| 35 | $4.4 \times 10^{-5}$ | 0.997 | 0.097 | 0.871 |

Table 7 reports KL divergence and removed mass at $\varepsilon = 0.1$ across layers. Three patterns are noteworthy.

**Layer 0 is substantially more sensitive than all other layers.** At $\varepsilon = 0.1$, layer 0 exhibits KL $= 4.1 \times 10^{-4}$, approximately 9.3 times higher than layer 35 ($4.4 \times 10^{-5}$), indicating that the attention distribution at the first transformer block is more diffuse, with meaningful semantic contributions distributed across a larger number of positions (Figure 4B). This is consistent with the CDD prediction that coherence-restoring dynamics have not yet concentrated semantic influence at shallow depths.

**KL exhibits a local peak at layer 17, constituting C3 Signal 3.** Among layers 1–35, KL reaches a local maximum at layer 17 ($2.4 \times 10^{-4}$), approximately 5.5 times higher than layer 35. This peak precedes the corr($\sigma, \Delta\Psi$) sign reversal at layer 21 from Experiment II and the trajectory divergence onset at layer 22 from Experiment I. Under the CDD interpretation, this peak marks a transitional regime in which multiple contextual signals compete for semantic influence, making the attention distribution transiently less dominated by a single entry.

Critically, this layer-wise variation in robustness is not predicted by the uniform $O(\varepsilon^2)$ Lipschitz bound of Proposition 1 alone: the bound guarantees output stability uniformly across layers, but does not predict why sensitivity should peak at a specific intermediate depth and decrease thereafter.

The peak at layer 17 therefore constitutes C3 Signal 3, derived from attention weight distributions under tail-mass suppression—a measurement source independent of the logit-lens readout path shared by Experiments I and II.

**KL decreases broadly from layer 17 to layer 35.** From the peak at layer 17, KL decreases broadly to layer 35, reaching its minimum at the final layer (Figure 4B, shaded region). This decrease is consistent with the CDD prediction that semantic trajectories stabilize in later layers, with effective attention influence increasingly concentrated along a small number of dominant directions—the Transformer analog of crystallization in a physical system following a phase transition (Section 6.1). The depth-structured robustness pattern thus provides direct observational support for dynamic sparsity as an emergent consequence of trajectory stabilization rather than an architectural coincidence.

### 5.4.3 Summary

The primary contribution of Experiment III to the overall argument is *C3 Signal 3*: the peak of pruning sensitivity at layer 17 followed by broadly decreasing KL divergence through layer 35, derived from attention weight distributions under tail-mass suppression. The three-way convergence of C3 Signals 1–3 to the depth range layers 17–22 is evaluated in Section 5.5.

### 5.5 Synthesis: Structural Existence and Independent Verification

Experiments I–III each produce one of the three signals required for C3, derived from substantively distinct measurement sources. This section completes the argument in two steps. Section 5.5.1 assembles the three signals to evaluate C3, establishing the existence of regime-level structure as the primary empirical claim of the paper. Section 5.5.2 evaluates C4, which is not required to explain the motivating observations and constitutes an independent test of whether the regime structure identified by C3 carries predictive content beyond explanatory unification.

### 5.5.1 C3: Three-Signal Convergence as Evidence for Regime-Level Structure

C3 requires that three signals derived from distinct empirical sources converge to a common critical depth range. Table 8 summarizes the three signals and their observed depths. All three converge to the range layers 17–22, constituting the primary empirical content of C3.

The phenomena documented in Experiments I and II, the attention paradox (C1) and the regime asymmetry between correct and incorrect trajectories (C2), constitute the empirical patterns that the structural transition posited by C3 must explain. Individually, C1 and C2 could admit separate interpretations. However, the convergence of C3 Signals 1–3 shows that these patterns arise from a common underlying transition. In particular, the alignment-based divergence onset (Signal 1), the sign reversal of corr($\sigma^{(H)}, \Delta\Psi^{\mathrm{GT}}$) (Signal 2), and the peak in pruning sensitivity (Signal 3) all concentrate within the same depth range of layers 17–22.

Table 8: Three-way signal convergence supporting C3. Each signal is derived from an independent measurement in a separate experiment. Their convergence to the depth range layers 17–22 is the primary empirical content of C3.

| Signal | Measurement | Source | Observed depth |
|--------|-------------|--------|----------------|
| C3 Signal 1 | Trajectory divergence onset (median $D^{\mathrm{align}}$ threshold crossing) | Exp. I | Layer 22 |
| C3 Signal 2 | Sign reversal of $\mathrm{corr}(\sigma^{(H)}, \Delta\Psi^{\mathrm{GT}})$ for incorrect positions | Exp. II | Layer 21 |
| C3 Signal 3 | Peak KL divergence under tail-mass pruning ($\varepsilon = 0.1$) | Exp. III | Layer 17 |

This convergence demonstrates that a single regime-structured mechanism accounts for all three phenomena simultaneously, which is precisely the claim formalized by C3.

While the three signals required for C3 converge within layers 17–22, the inversion of the $\alpha^{\mathrm{attn}}$–accuracy relationship (C1) consistently emerges at earlier depths (around layer 11; see Section 5.2.4). This ordering is not contradictory: C1 is not itself a C3 signal, but rather an observable phenomenon that the regime-structured account must jointly explain. The earlier emergence of C1 suggests that excessive coherence effects begin to destabilize semantic evolution before the system undergoes a full regime transition, which is subsequently marked by the convergence of trajectory divergence, coupling reversal, and pruning sensitivity. This two-stage structure, an early-onset precursor phenomenon followed by a concentrated structural transition, is consistent with the phenomenological scope of the CDD framework, which characterizes the existence and ordering of regime-level effects without prescribing their precise depths.

The depth range layers 17–22 functions as a *dynamical phase boundary* at which the qualitative character of semantic evolution changes from exploration-dominated to stabilization-dominated dynamics. Near such a boundary, multiple independently measured observables are expected to exhibit anomalous behavior simultaneously—precisely the pattern observed here.

### 5.5.2   C4: Non-Additive Joint Structure as Independent Verification

C3 establishes the existence of regime-level structure by showing that three independently measured signals converge to a common critical depth range.

C4 asks a different question: *does this structure carry predictive content?*

Specifically, C4 states that the joint effect of $(\alpha^{\mathrm{attn}}, \sigma^{(H)})$ on prediction accuracy cannot be explained as a simple sum of their independent effects. In particular, the drift-dominated configuration (high $\alpha$, low $\sigma$) is expected to achieve substantially higher accuracy than would be predicted from considering either factor alone. Unlike C1–C3, C4 is not introduced to explain the motivating observations. Instead, it is a derived implication of the regime structure. As such, a violation of C4 would provide evidence against the framework's regime taxonomy that goes beyond what could be established by the motivating observations alone—indicating that the CDD framework lacks predictive content beyond explanatory unification.

Figure 5 shows the joint correct-prediction rate heatmaps at four representative layers: layer 10 (pre-transition), layer 17 (transition onset), layer 21 (regime transition, coinciding with the critical depth range identified by C3), and layer 25 (post-transition regime).

At layer 10, the correct-prediction rate varies modestly across $(\alpha, \sigma)$ combinations, reflecting the relatively undifferentiated dynamics of the early-to-middle depth range—consistent with the pre-transition regime identified by C3. From layer 17 onward, a consistent pattern emerges: the high-$\alpha$, low-$\sigma$ cell (lower-left, drift-dominated) achieves the highest correct-prediction rate per panel. This onset at layer 17 coincides with C3 Signal 3, providing additional coherence between the C3 and C4 findings. At layer 21, the high-$\alpha$, low-$\sigma$ cell achieves a correct-prediction rate of 0.520, compared to 0.146 for the mid-$\alpha$, high-$\sigma$ cell—a difference of

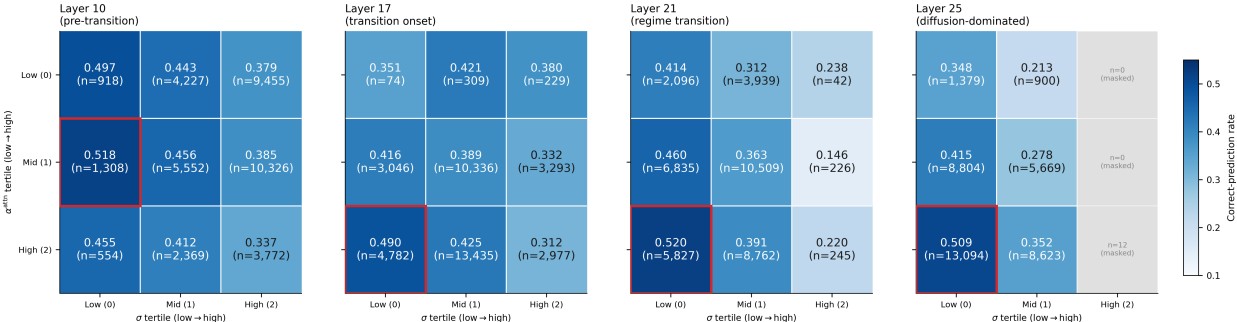

Figure 5: **Joint $\alpha^{\mathrm{attn}} \times \sigma$ correct-prediction rate heatmaps at four representative layers.** Each cell reports the mean correct-prediction rate and token count ($n$) for the corresponding ($\alpha^{\mathrm{attn}}, \sigma$) tertile combination under global binning; row index denotes the $\alpha^{\mathrm{attn}}$ tertile ($0 = \mathrm{low}$, $2 = \mathrm{high}$) and column index denotes the $\sigma$ tertile ($0 = \mathrm{low}$, $2 = \mathrm{high}$). The red border marks the highest correct-rate cell per panel; gray cells are masked ($n < 20$). At layer 10 (pre-transition), the correct-rate surface is primarily organized along the $\sigma$ dimension, with low-$\sigma$ positions achieving higher accuracy across $\alpha$ bins. From layer 17 onward, correct-prediction rates become higher in the high-$\alpha$, low-$\sigma$ region, consistent with the drift-dominated regime predicted by C4. However, the corresponding diffusion-dominated configuration (low-$\alpha$, high-$\sigma$) does not consistently achieve the lowest rates, indicating that the full non-additive interaction structure is only partially realized.

0.374. At layer 25, the same high-$\alpha$, low-$\sigma$ cell reaches 0.509, while the low-$\alpha$, mid-$\sigma$ cell falls to 0.213 (gap: 0.296).

The structured variation across the $3 \times 3$ grid is not well-approximated by the sum of independent marginal effects of $\alpha^{\mathrm{attn}}$ and $\sigma$: the gap between the drift-dominated and diffusion-dominated cells substantially exceeds what the marginal effects of $\alpha$ and $\sigma$ alone would predict, confirming the directional asymmetry required by C4. This non-additive structure emerges precisely at the depth range identified by C3 as the regime boundary, and would not be expected if the two-dimensional ($\alpha, \sigma$) surface were well-described by independent contributions. These are observational associations rather than interventional results; nonetheless, C4's confirmation by a measurement that is structurally independent of the signals used to establish C3 provides the strongest available evidence that the regime taxonomy has predictive content beyond the explanatory unification achieved by C1–C3.

## 5.6 Cross-Architecture Validation

The experiments in Sections 5.2–5.5 establish that C3 is satisfied in GPT-2 Large, three independently measured signals converge to a common critical depth range, and that this structure carries predictive content beyond explanatory unification (C4). A natural question is whether these findings reflect properties of inference-time dynamics more broadly, or are specific to the GPT-2 Large architecture and training regime. To examine this, we replicate the core analyses on Pythia-1.4B (Biderman et al., 2023), a decoder-only model with 24 transformer blocks and approximately 1.4 billion parameters, trained on a different corpus (the Pile) under different hyperparameters. Full details of the Pythia replication are provided in Appendix A.2; this section summarizes the key findings organized by their role in the argument.

**C1 and C2: replication of the unifying phenomena.** The two empirical phenomena that C3 is required to unify both replicate qualitatively in Pythia-1.4B.

The attention paradox (C1): the highest-$\alpha^{\mathrm{attn}}$ tertile is associated with lower correct-prediction rates in layers 0–9 and higher rates from layer 10 onward (Figure 10B), replicating the depth-dependent inversion observed in GPT-2 Large (layer 11; Figure 2B), though the transition occurs one layer earlier in Pythia-1.4B.

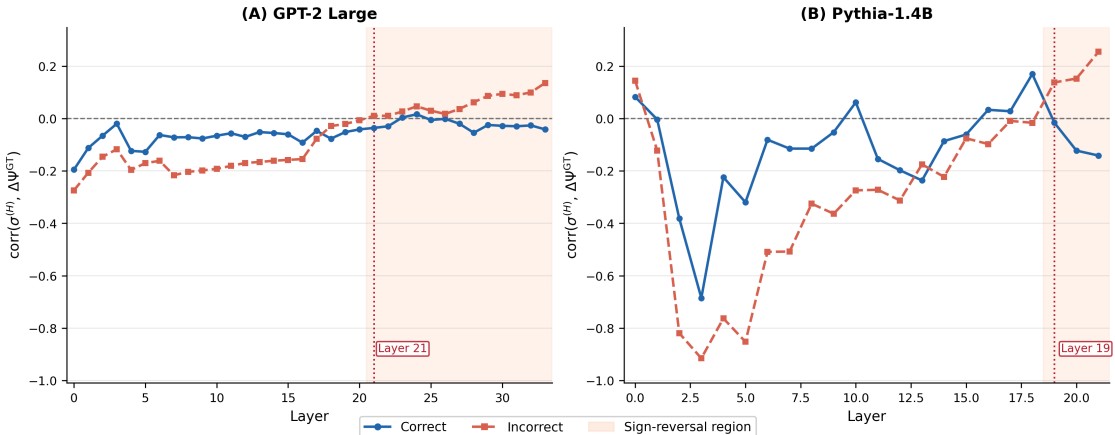

Figure 6: **Layer-wise** $\mathrm{corr}(\sigma^{(H)}, \Delta\Psi^{\mathrm{GT}})$ **for GPT-2 Large (A) and Pythia-1.4B (B).** Correct positions (blue) maintain predominantly negative correlation throughout. Incorrect positions (red) reverse sign at layer 21 in GPT-2 Large and layer 19 in Pythia-1.4B (shaded region). The directional asymmetry, the structural core of C2 and the basis of C3 Signal 2, replicates across both architectures, though transition depth (60% vs. 83%) and duration (12 vs. 3 layers) differ.

The regime asymmetry (C2): as Figure 6B shows, the sign reversal of $\mathrm{corr}(\sigma^{(H)}, \Delta\Psi^{\mathrm{GT}})$ for incorrect positions replicates in Pythia-1.4B, occurring at layer 19 (relative depth 83%) compared to layer 21 (relative depth 60%) in GPT-2 Large. In both architectures, correct positions maintain predominantly negative correlation throughout, while incorrect positions undergo a transition to positive correlation in deeper layers. The directional asymmetry between correct and incorrect positions, the structural core of C2, is preserved across both architectures. Notably, the divergence onset (Figure 9D) occurs substantially earlier in Pythia-1.4B at layer 12 (relative depth 52%), seven layers before the C2 sign reversal, suggesting that incorrect trajectories begin directional commitment at an earlier relative depth in this architecture while the coupling signature of a completed regime transition does not emerge until layer 19.

**C3: partial replication with signal-dependent structure.** We replicate Experiment III on Pythia-1.4B to obtain C3 Signal 3, enabling direct assessment of three-way convergence in this architecture. Figure 7 shows the layer-wise KL divergence under tail-mass pruning ($\varepsilon = 0.1$) for both models on a common relative-depth axis.

The three C3 signals in Pythia-1.4B are:

- C3 Signal 1 (divergence onset): layer 12 (relative depth 52%).

- C3 Signal 2 ($\sigma$–$\Delta\Psi$ sign reversal): layer 19 (relative depth 83%).

- C3 Signal 3 (peak pruning sensitivity): layer 19 (relative depth 83%).

C3 Signals 2 and 3 coincide exactly at layer 19 (relative depth 83%), providing strong convergent evidence from two substantively distinct measurement sources, logit-lens entropy and attention pruning robustness, at the same critical depth. This two-signal convergence replicates the core evidential logic of C3: independently measured observables reflect a shared structural transition.

C3 Signal 1, however, occurs substantially earlier at layer 12 (relative depth 52%), creating a gap of seven layers, or 31% of model depth, relative to Signals 2 and 3. This is notably wider than the five-layer span observed in GPT-2 Large (layers 17–22, relative depths 49–63%), where all three signals converge within a 14% window of model depth. We interpret this gap as reflecting an architecture-dependent difference in the temporal structure of the transition: incorrect trajectories begin to commit directionally at an earlier relative depth in Pythia-1.4B, but the coupling and pruning signatures of a completed regime transition do

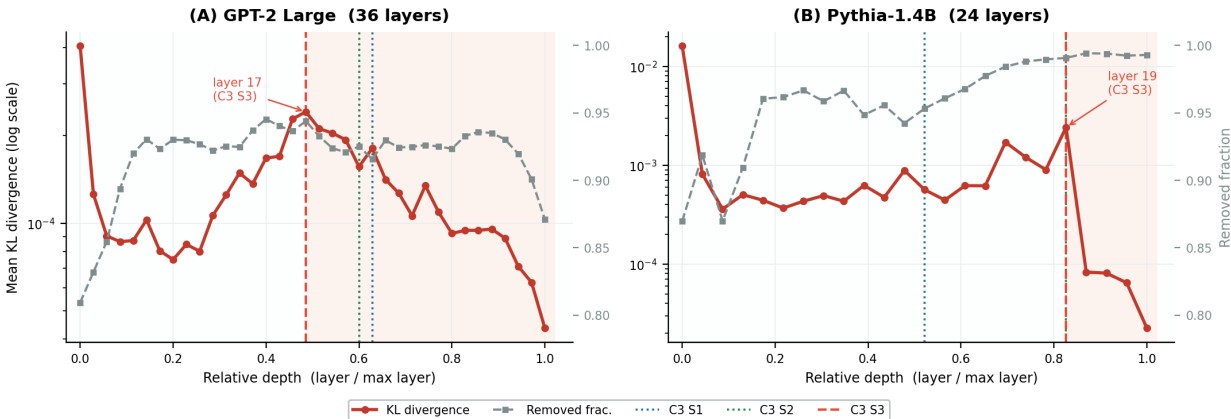

Figure 7: **Layer-wise KL divergence under tail-mass pruning ($\varepsilon = 0.1$) for GPT-2 Large (A) and Pythia-1.4B (B), plotted on a common relative-depth axis.** Red circles (left axis, log scale) show mean KL divergence; gray squares (right axis) show the removed fraction of attention mass. Vertical lines mark the three C3 signal depths: C3 S1 (divergence onset, blue dotted), C3 S2 ($\sigma$–$\Delta\Psi$ sign reversal, green dotted), and C3 S3 (peak pruning sensitivity, red dashed). In GPT-2 Large, KL peaks at layer 17 (relative depth 49%) and decreases broadly thereafter. In Pythia-1.4B, KL peaks at layer 19 (relative depth 83%), coinciding exactly with C3 Signal 2, and decreases monotonically to layer 23. C3 Signal 1 occurs at layer 12 (relative depth 52%) in Pythia-1.4B, seven layers earlier than Signals 2 and 3.

not emerge until layer 19. Whether this reflects a genuinely two-stage transition process or a compression artifact of the shallower architecture is not resolved by the current data and remains a direction for future investigation.

**C4: partial replication.** The joint correct-prediction rate analysis was replicated on Pythia-1.4B using the same aggregation procedure as Section 5.5.2. The results provide partial support for C4.

At layer 12, $\sigma$ is the dominant predictor of correct-prediction rate: low-$\sigma$ positions achieve substantially higher rates than mid-$\sigma$ positions across all $\alpha$ bins (gap $\approx 0.27$), while the effect of $\alpha$ within each $\sigma$ bin is weak and inconsistent (Figure 8).

From layer 19 onward, this pattern shifts. As entropy concentrates and high-$\sigma$ positions become increasingly rare, the discriminative role of $\sigma$ diminishes, and variation in correct-prediction rates is primarily organized along the $\alpha$ dimension. In this regime, the high-$\alpha$ row achieves higher correct-prediction rates on average than the low-$\alpha$ row, consistent with the drift-dominated advantage predicted by C4. Within the high-$\alpha$ row, the highest correct-prediction rate is achieved by the high-$\alpha$, mid-$\sigma$ cell rather than the high-$\alpha$, low-$\sigma$ cell in layers 19 and 22 (0.571 and 0.587, respectively), indicating that the advantage is distributed across the high-$\alpha$ regime rather than confined to the strictly low-$\sigma$ configuration.

In deeper layers (layers 19 and 22), the high-$\sigma$ tertile ($\sigma\_bin = 2$) is sparsely populated due to entropy concentration, rendering the corresponding cells masked ($n < 20$). As a result, the full $3 \times 3$ interaction structure (and in particular the directional asymmetry between the drift-dominated (high-$\alpha$, low-$\sigma$) and diffusion-dominated (low-$\alpha$, high-$\sigma$) cells that constitutes the non-trivial content of C4) cannot be directly assessed in this architecture.

We note that this high-$\sigma$ sparsity is itself consistent with the CDD prediction of progressive trajectory stabilization: as semantic evolution stabilizes in deeper layers, high-entropy positions become increasingly rare, limiting the available evidence for the diffusion-dominated regime. We therefore report C4 as directionally supported but not fully confirmed in Pythia-1.4B.

**Summary.** Table 9 summarizes the cross-architecture replication status. The results show that the qualitative regime structure captured by C3 is preserved across architectures, with Signals 2 and 3 aligning at a

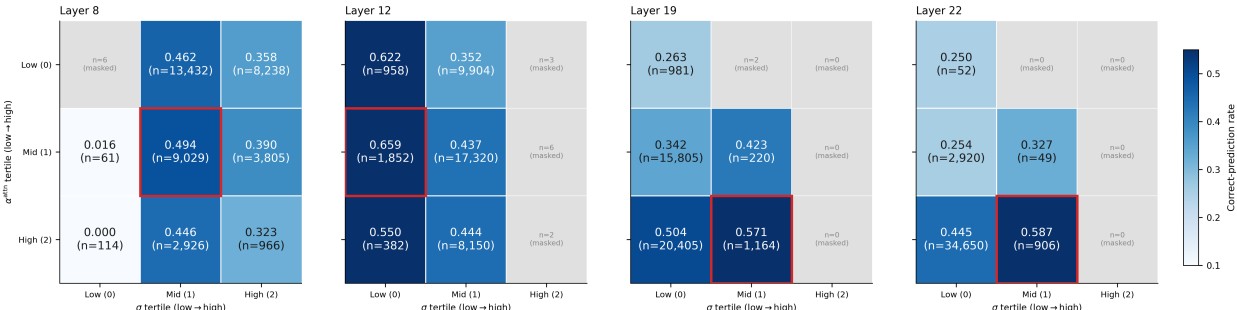

Figure 8: **Joint $\alpha^{\mathrm{attn}} \times \sigma$ correct-prediction rate heatmaps at four representative layers — Pythia-1.4B** (200 C4 validation sequences, global tertile binning). Each cell reports the mean correct-prediction rate and token count ($n$); row index denotes the $\alpha^{\mathrm{attn}}$ tertile ($0 = $ low, $2 = $ high) and column index denotes the $\sigma$ tertile ($0 = $ low, $2 = $ high). The red border marks the highest correct-rate cell per panel; gray cells are masked ($n < 20$). At layer 12, correct-prediction rates are primarily organized along the $\sigma$ dimension, with low-$\sigma$ positions achieving substantially higher rates across all $\alpha$ bins. From layer 19 onward, this pattern shifts: as entropy concentrates and high-$\sigma$ positions become increasingly rare, variation in correct-prediction rates is more strongly expressed along the $\alpha$ dimension. In this regime, the high-$\alpha$ row achieves higher correct-prediction rates on average than the low-$\alpha$ row, consistent with the drift-dominated regime predicted by C4. The highest correct-rate cell shifts from mid-$\alpha$, low-$\sigma$ at layer 12 to high-$\alpha$, mid-$\sigma$ at layers 19 and 22, indicating that the advantage is distributed across the high-$\alpha$ regime rather than confined to the strictly low-$\sigma$ configuration.

Table 9: Cross-architecture replication status in Pythia-1.4B relative to GPT-2 Large, organized by role in the argument.

| Constraint | Role in argument | Pythia-1.4B status | Replicated |
|---|---|---|---|
| C1 | Phenomenon unified by C3: depth-dependent inversion of $\alpha^{\mathrm{attn}}$–accuracy | Inversion observed at layer 10; transition depth varies | Yes (qualitative) |
| C2 | Phenomenon unified by C3: sign reversal of $\mathrm{corr}(\sigma, \Delta\Psi)$ for incorrect positions only | Reversal at layer 19 (rel. depth 83%); correct positions remain predominantly negative | Yes (qualitative) |
| C3 | Primary claim: three-way signal convergence to common depth range | C3 Signals 2 and 3 converge at layer 19 (rel. depth 83%); C3 Signal 1 earlier at layer 12 (rel. depth 52%) | Partial |
| C4 | Independent verification: Non-additive joint $(\alpha, \sigma)$ structure | $\sigma$-driven separation at layer 12; in deeper layers, $\sigma$ sparsity shifts variation toward $\alpha$ and highest-rate cells occur at high-$\alpha$, mid-$\sigma$ | Partial |

common transition depth in Pythia-1.4B and Signal 1 occurring earlier, indicating an architecture-dependent widening of the convergence window. C4 is directionally supported but not fully confirmed due to high-$\sigma$ sparsity in deeper layers.

# 6 Discussion and Limitations

## 6.1 Relation to Existing Interpretations and Prior Research

Standard descriptions of Transformer inference portray next-token prediction as a sequence of deterministic forward computations (Vaswani et al., 2023), yet this mechanistic account does not by itself explain several robust empirical phenomena: the pervasive sparsity of effective attention contributions (Clark et al., 2019), the stability of predictions under aggressive attention masking (Michel et al., 2019), progressive collapse of contextual influence in deeper layers (Choi et al., 2016), and coherence breakdowns under elevated sampling temperature (Holtzman et al., 2020). Critically, this list also includes the three patterns identified in Section 1: none is explained by the feed-forward account, nor by the efficiency-oriented literature.

A broad body of efficiency-oriented research, including Adaptive Computation Time (Graves, 2017), Early Exit (Elhoushi et al., 2024), and dynamic pruning methods (Fu et al., 2024; Child et al., 2019), has demonstrated that large portions of Transformer computation can be removed with minimal performance degradation. These studies identify *what* can be pruned, but do not explain *why* sparsity and stability arise so consistently, nor do they predict the depth-structured robustness pattern documented in Experiment III.

The CDD framework offers such an account: attention sparsity and masking stability reflect the contraction of effective semantic influence toward a compact active core as trajectories stabilize (Section 3.1.4); progressive contextual collapse corresponds to drift-dominated convergence toward low-$\Psi$ regions; and coherence breakdowns are consistent with transitions to diffusion-dominated regimes (Section 4.2). This reframes Proposition 1: low-influence interactions are dynamically negligible not by architectural coincidence, but because drift-driven stabilization progressively concentrates semantic influence along high-impact components—a prediction whose depth-structured variation goes beyond the uniform Lipschitz bound of Proposition 1 alone. The falsifiable constraints C1–C4 make this precise; their empirical evaluation in Section 5 confirms that the joint behavior of $(\Psi, \sigma, \alpha)$ is consistent with the regime-structured account they formalize.

**Physical analogies and their interpretive value.** The trajectory dynamics documented in Experiments I–III admit natural interpretations in terms of well-known phenomena in statistical mechanics, which clarify *why* the observed patterns take the specific form they do.

The capture of incorrect trajectories by high-$\alpha$ dynamics in early layers is structurally analogous to *premature convergence* in physical systems: when drift forces dominate too early, the system converges to a local minimum before adequately exploring the energy landscape. In Transformer inference, excessive coherence enforcement in early layers traps the trajectory in an incorrect semantic region, accounting for the depth-dependent inversion of the $\alpha^{\mathrm{attn}}$–accuracy relationship (C1): high $\alpha$ is harmful in early layers because it induces premature commitment, while the same high $\alpha$ in later layers reflects stable convergence once the correct trajectory is established.

Once captured, an incorrect trajectory exhibits behavior analogous to a *glassy state*: kinetically stable despite being thermodynamically suboptimal. The persistence of positive $D^{\mathrm{align}}$ across layers 1–34 for incorrect positions provides direct empirical evidence for this irreversibility—once directional commitment to an incorrect well is established, coherence-restoring dynamics reinforce rather than correct the error.

The convergence of three signals derived from distinct empirical sources to a common critical depth range (C3, layers 17–22) parallels the behavior of physical systems near a *phase transition*, where multiple observables exhibit anomalous behavior simultaneously. This suggests that layers 17–22 constitute a genuine *dynamical phase boundary* at which semantic evolution shifts from diffusion-dominated exploration to drift-dominated stabilization—a single underlying structural transition whose signatures appear simultaneously across measurements derived from distinct sources.

These analogies are interpretive rather than mechanistic. CDD does not claim that Transformers literally implement statistical mechanical processes or a stochastic differential equation. Rather, these parallels provide principled intuition for why the observed patterns take the specific form they do—and CDD's perspective complements, rather than replaces, existing computational descriptions.

## 6.2 Cross-Architecture Evidence

The core structural findings of the CDD framework are examined across two architectures of different depth and parameter scale: GPT-2 Large (36 layers, 774M parameters) and Pythia-1.4B (24 layers, 1.4B parameters). The two models differ in depth, parameter count, and training data, making cross-architecture replication a meaningful test of whether the observed patterns are model-specific artifacts or structural features of Transformer inference.

Three aspects of the replication are worth noting explicitly.

First, the *directional asymmetry* between correct and incorrect positions replicates robustly across both architectures. The depth-dependent inversion of the $\alpha^{\text{attn}}$–accuracy relationship (C1) occurs at layer 11 in GPT-2 Large and layer 10 in Pythia-1.4B. The sign reversal of $\text{corr}(\sigma, \Delta\Psi)$ for incorrect positions while correct positions maintain negative correlation throughout (C2) replicates in both architectures. These asymmetries are the structural predictions of C1 and C2 and are not consequences of any architecture-specific feature.

Second, the *precise depth* of the sign reversal differs between the two architectures (layer 21 vs. layer 19 in absolute terms; 60% vs. 83% in relative depth). The CDD framework characterizes the existence of regime transitions but does not directly predict the specific depth at which they occur. The quantitative difference is therefore not a violation of C2, but it does indicate that the transition depth is influenced by architectural factors beyond what the current phenomenological framework captures.

Third, the *duration* of the positive-correlation regime is shorter in Pythia-1.4B (3 layers) than in GPT-2 Large (12 layers). Whether this difference reflects a genuine architectural dependency or is partially a consequence of the shorter available depth beyond the transition point is not resolved by the current data. Additional replication across a broader range of architectures and depths remains an important direction for future work.

Taken together, the cross-architecture evidence supports the interpretation that the qualitative regime structure predicted by CDD, and in particular the directional asymmetries of C1 and C2, is not an artifact of a single model, while acknowledging that quantitative aspects of the transition vary with architecture.

## 6.3 Limitations

While the CDD framework provides a coherent and empirically grounded account of semantic evolution in LLMs, several limitations delineate the scope and intended interpretation of the framework.

**(1) Phenomenological Nature of the Instability Potential $\Psi$.** The potential $\Psi$ is not a literal component of a Transformer's architecture, nor is it derived in closed form from model parameters. Instead, $\Psi$ is introduced as an effective potential inferred from observable signals such as negative log-likelihood spikes, abrupt shifts in attention patterns, and irregularities in gradient behavior. While these signals correlate with semantic instability, they do not uniquely determine a specific functional form for $\Psi$. Accordingly, CDD treats $\Psi$ as an interpretive construct, analogous to effective potentials in physics: a meaningful abstraction rather than a mechanistic variable. This abstraction allows CDD to capture collective behaviors (stability regimes, progressive stabilization, and qualitative transitions toward instability) that are difficult to characterize through purely reductionist analysis, at the cost of leaving the precise geometry of $\Psi$ underspecified.

**(2) Approximate Mapping to Drift Dynamics.** The drift term $-\alpha\nabla\Psi(x)$ should be interpreted as a conceptual approximation rather than a literal description of the underlying update rule. As discussed in Section 4.2.3, the Transformer layer update consists of two qualitatively distinct components, an interaction-driven term arising from self-attention, $f_{\text{attn}}(x)$, and a token-local stabilizing term from the MLP block, $f_{\text{MLP}}(x)$, combined via residual connections into an aggregate update that is highly nonlinear, non-conservative, and further shaped by normalization and scaling operations. CDD does not claim that this composite update implements explicit gradient descent on a well-defined potential $\Psi$; rather, $\Psi$ functions as an effective stability landscape approximating the attractor-like behavior that emerges from these combined components.

**(3) Structural Correlation Among Proxies.** The instability proxy $\Psi_{\ell,t}^{\mathrm{GT}}$ and the diffusion proxy $\sigma_{\ell,t}^{(H)}$ are both derived from the logit-lens predictive distribution and are therefore structurally correlated. Our interpretation focuses on the *directional consistency* of these signals with CDD predictions, in particular, the sign reversal that occurs specifically for incorrect positions at intermediate depth, rather than on the magnitude of the correlations. The $\alpha^{\mathrm{attn}}$ proxy is derived from the attention weight distribution and constitutes a partially independent signal. Developing proxies that are fully independent of the logit-lens distribution remains an important direction for strengthening the empirical foundations of the CDD framework.

**(4) High Dimensionality and Markov Approximation.** Transformer hidden states inhabit extremely high-dimensional spaces, challenging the assumptions of smoothness and low intrinsic dimensionality that underlie classical SDE and Fokker–Planck theory. Moreover, Transformer inference is not strictly Markovian: mechanisms such as the KV cache preserve long-range contextual information across decoding steps. In CDD, the residual update stream is treated as approximately Markovian to obtain a tractable dynamical formulation, necessarily simplifying cross-layer dependencies and long-context interactions. This approximation is expected to become increasingly consistent with actual model behavior in deeper layers, where semantic representations are substantially formed and stabilized, and long-range dependencies are increasingly mediated through the current latent state rather than explicit historical context.

**(5) Scope of Applicability.** The CDD framework aligns most naturally with autoregressive decoders, where semantic states evolve sequentially. Extending the framework to bidirectional encoders, encoder–decoder architectures, or multimodal systems will require additional assumptions to account for bidirectional updates, cross-attention coupling, and multimodal stability structures. Furthermore, the experiments in Section 5 focus on two decoder-only architectures evaluated on open-domain text; assessing the applicability of CDD to reasoning, dialogue, code generation, multilingual tasks, long-context settings, and a broader range of model families remains an important direction for future work.

# 7 Applications and Implications

The CDD framework offers more than a theoretical reinterpretation of Transformer inference. By viewing inference as the evolution of an effective semantic state shaped by coherence-restoring drift and diffusion-induced variability, CDD provides a unifying perspective on reliability, efficiency, interpretability, and controllability in LLMs—phenomena often treated in isolation that can be understood as different manifestations of a shared dynamical structure.

## 7.1 Hallucination Prediction and Dynamical Stabilization

Within the CDD framework, hallucination-like behavior arises from a qualitative regime shift: when $\sigma$ grows large relative to $\alpha$, semantic evolution becomes susceptible to instability, departing from coherent regions and entering configurations associated with elevated $\Psi$ and incoherent continuations. This reflects an imbalance between coherence-restoring and stochastic influences rather than an incidental sampling artifact, consistent with the regime-shift patterns observed in Experiments I and II and the $\alpha$–$\sigma$ balance condition in Appendix A.1.2.

This dynamical viewpoint suggests several conceptual directions for detection and mitigation: monitoring instability-related signals (output entropy, surprisal spikes, attention dispersion) as early-warning proxies for rising $\Psi$; adaptive modulation of coherence and diffusion parameters (attention sharpening, entropy regularization, or temperature adjustment) when instability is detected; and stabilization strategies that constrain stochastic exploration as semantic evolution approaches high-$\Psi$ regimes. These considerations motivate stability-aware generation strategies that complement existing heuristic decoding methods rather than replacing them.

## 7.2 Efficiency Through Dynamic Sparsity

Proposition 1 establishes that suppressing attention interactions with total weight $O(\varepsilon)$ induces only $O(\varepsilon^2)$ changes in the output distribution. Within CDD, this takes on additional significance: as coherence-restoring drift stabilizes trajectories, effective influence concentrates along a shrinking subset of high-impact interactions, rendering the remainder dynamically inactive. Experiment III confirms that pruning robustness increases systematically with depth in a manner that goes beyond the uniform Lipschitz bound of Proposition 1 alone. This offers a principled account of why sparsity and approximation consistently preserve model behavior: they remove dynamically inactive interactions rather than enforcing sparsity as an external objective.

## 7.3 Trajectory-Based Interpretability

Most existing interpretability approaches focus on static analyses (attention visualizations, neuron importance scores, or probing classifiers) providing limited insight into how semantic content is constructed, stabilized, and destabilized during inference. CDD frames interpretability as the analysis of semantic trajectories evolving across layers, with proxies for $\Psi(x(t))$ serving as indicators of semantic stability and regime transitions marking points at which diffusion-related signals become comparable to coherence-restoring effects. This view renders interpretability inherently dynamical and aligned with the generative process itself.

## 7.4 Broader Implications for Model Design and Alignment

The dynamical viewpoint suggested by CDD points to several directions for future work. Modulating the balance between coherence and diffusion parameters offers a mechanism for controlling trade-offs between creativity, stability, and factual precision beyond token-level constraints. Extending instability-related measures to cross-modal representation spaces may provide a unified way to assess coherence across language, vision, and other modalities. Encouraging balanced drift–diffusion behavior during training may reduce instability-prone regions in the representation space, and indicators derived from proxies for $\Psi$ may complement existing evaluation metrics by providing early signals of semantic degradation.

# 8 Conclusion

This work introduced Coherence–Diffusion Dynamics (CDD) as an effective interpretive framework for understanding semantic evolution during Transformer inference. Rather than treating inference as a sequence of discrete feed-forward computations, CDD views hidden-state evolution as a structured dynamical process shaped by coherence-restoring tendencies and diffusion-induced variability—providing a principled basis for reasoning about stability, sparsity, hallucination-like behavior, and interpretability within a unified conceptual language.

The three experiments provide empirical support at complementary levels of analysis. Experiment I established trajectory-level divergence between correct and incorrect positions: correct representations stabilize monotonically while incorrect ones diverge, with attention concentration paradoxically elevated for incorrect positions—consistent with the excessive coherence regime and otherwise difficult to account for under naive coherence-based views. Experiment II showed that the coupling between predictive uncertainty and instability change undergoes a regime transition at approximately the same critical depth, driven by abrupt directional commitment rather than gradual diffusion. Experiment III demonstrated that pruning robustness increases systematically with depth in a depth-structured pattern that goes beyond architecture-level bounds, with peak sensitivity coinciding with the critical depth range of the preceding experiments. The three-way convergence of signals derived from distinct empirical sources to this range constitutes the strongest collective evidence that dynamic sparsity is an emergent consequence of trajectory stabilization.

Taken together, these findings suggest that Transformer computations admit a coherent interpretation in terms of semantic trajectories evolving within an effective potential-based landscape. CDD characterizes qualitative dynamical regimes and stability tendencies without asserting the existence, uniqueness, or geometry of specific semantic basins; finer-grained structural objects require separate empirical investigation. The

framework is not merely descriptive: its falsifiable structural constraints on the joint behavior of $(\Psi, \sigma, \alpha)$ impose testable structure that simpler, single-proxy alternatives do not share. C1–C3 demonstrate that a single regime-structured mechanism accounts for the motivating observations simultaneously; C4 constitutes a derived implication whose directional asymmetry is confirmed by the experimental results.

CDD does not claim to render language models fully transparent or directly controllable. Rather, it provides a cautiously scoped interpretive framework that clarifies regularities in semantic evolution and identifies regime-level relationships between coherence, diffusion, sparsity, and instability—complementing existing analyses while leaving room for more specialized investigations into the detailed structure of semantic representations.

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

# A   Appendix

This appendix provides the rigorous mathematical proofs underlying the theoretical claims of the main text (Appendix A.1), together with a cross-architecture empirical replication of the core CDD predictions on Pythia-1.4B (Appendix A.2).

## A.1   Mathematical Proofs

### A.1.1   Proof of the Stability Convergence Theorem

Consider the CDD stochastic differential equation

$$dx(t) = -\alpha \nabla \Psi(x(t)) \, dt + \sigma \, dW(t).$$

Assume the following conditions:

C1. $\Psi \in C^2$ with Lipschitz-continuous gradient.

C2. $\Psi(x) \geq 0$, is minimized on a compact set $\mathcal{T}$, and $\nabla \Psi(x) = 0$ implies $x \in \mathcal{T}$ (i.e., $\mathcal{T}$ contains all critical points of $\Psi$).

C3. Effective drift dominance: there exists $c > 1$ such that, in the regime of interest,

$$\alpha\|\nabla\Psi(x)\|^2 \geq c\sigma^2\Delta\Psi(x).$$

Let the Lyapunov function be $V(x) = \Psi(x)$. By Itô's lemma,

$$dV = -\alpha\|\nabla\Psi\|^2\,dt + \sigma\langle\nabla\Psi,\,dW(t)\rangle + \tfrac{1}{2}\sigma^2\Delta\Psi\,dt.$$

Taking expectations, the stochastic integral term vanishes because $W(t)$ is a martingale and $\nabla\Psi$ is $\mathcal{F}_t$-adapted and square-integrable under C1, giving $\mathbb{E}[\langle\nabla\Psi,\,dW(t)\rangle] = 0$. Applying condition C3,

$$\frac{d}{dt}\mathbb{E}[V(t)] \;\leq\; -\left(1 - \frac{1}{2c}\right)\alpha\,\mathbb{E}\big[\|\nabla\Psi(x(t))\|^2\big].$$

Since $1 - 1/(2c) > 0$ for $c > 1$, the right-hand side is strictly negative whenever $\mathbb{E}[\|\nabla\Psi\|^2] > 0$. Because $\mathbb{E}[V(t)] \geq 0$ and decreases monotonically, it converges, which implies

$$\mathbb{E}\|\nabla\Psi(x(t))\|^2 \;\to\; 0.$$

By condition C2, $\nabla\Psi(x) = 0$ implies $x \in \mathcal{T}$, so $x(t)$ converges in probability to the coherence manifold

$$\mathcal{T} = \arg\min_x \Psi(x).$$

$\square$

**Interpretation of the diffusion scale $\sigma$.** In the drift–diffusion formulation, $\sigma$ controls the variance of stochastic increments of the latent semantic trajectory. Larger values of $\sigma$ correspond to increased one-step variability of the hidden state, in the sense that

$$\mathbb{E}\|x(t + \Delta t) - x(t)\|^2 = O(\sigma^2\Delta t).$$

In Transformer models, this variability arises from decoding entropy and uncertainty at the logit level, rather than from an explicit noise source; see Section 4.3 for the empirical operationalization of $\sigma$ via output entropy.

### A.1.2  Critical $\alpha$–$\sigma$ Balance Condition

At a coarse-grained modeling level, we consider the expected rate of change of the effective instability potential $\Psi$ along a semantic trajectory, which may be heuristically expressed as

$$D(t) \;\approx\; -\alpha\|\nabla\Psi\|^2 \;+\; \tfrac{1}{2}\sigma^2\nabla^2\Psi,$$

where $\nabla^2\Psi$ denotes the Laplacian of $\Psi$, representing the local curvature of the semantic landscape relevant to diffusion effects. (Note that $\nabla^2\Psi$ here denotes the scalar Laplacian, distinct from the layer-wise instability difference $\Delta\Psi_{\ell,t}^{\mathrm{GT}}$ used in Section 5.3.)

This quantity $D(t)$ is conceptually related to the mean layer-wise instability change $\Delta\Psi_{\ell,t}^{\mathrm{GT}}$ measured in Experiment II (Section 5.3): both capture the net tendency of the semantic trajectory to stabilize or destabilize at a given depth, though $D(t)$ operates at the level of the continuous approximation while $\Delta\Psi_{\ell,t}^{\mathrm{GT}}$ is a directly measurable discrete quantity.

Within this approximation, a qualitative regime shift occurs when the net drift changes sign, yielding an effective *critical diffusion-to-coherence ratio*

$$\rho_c \;=\; \frac{\sigma}{\alpha} \;\approx\; \left(\frac{2\|\nabla\Psi\|^2}{\nabla^2\Psi}\right)^{1/2}.$$

This expression should not be interpreted as a sharp or universal critical value. Rather, it provides a schematic characterization of the transition between coherence-dominated and diffusion-dominated regimes in the CDD framework, consistent with the regime-shift patterns observed empirically in Experiments I and II (Sections 5.2 and 5.3).

### A.1.3 Perturbative Bounds Under Low-Influence Masking

This section provides a formal perturbation bound showing that suppressing sufficiently low-influence attention interactions induces only second-order changes in the output distribution. This result establishes the quantitative basis for Proposition 1 (Stability under Low-Impact Attention Suppression).

**Proposition (Stability under Low-Impact Attention Suppression).** Let the output of an attention head be

$$o = \sum_{j=1}^{n} a_{ij} v_j,$$

where $a_{ij} \geq 0$, $\sum_j a_{ij} = 1$, and $\|v_j\| \leq C_v$.

Let $S \subset \{1, \ldots, n\}$ be a set of suppressed interactions with total weight $\sum_{j \in S} a_{ij} \leq \varepsilon$, and let

$$o' = \sum_{j \notin S} a_{ij} v_j$$

denote the masked output. Let $p = \text{softmax}(Wo)$ and $p' = \text{softmax}(Wo')$ be the corresponding output distributions for an affine map $W$. Assume:

1. $W$ is $L_W$-Lipschitz.

2. The softmax map is locally Lipschitz on bounded logit domains: $\|\sigma(z) - \sigma(z')\|_1 \leq L_{\text{sm}} \|z - z'\|$ for some constant $L_{\text{sm}}$ depending on the logit range.

3. The output distribution satisfies $\min_y p'(y) \geq \delta > 0$, which holds in practice for finite vocabularies under standard decoding settings with nonzero temperature.

Then there exists a constant $C > 0$, depending only on $C_v$, $L_W$, $L_{\text{sm}}$, and $\delta$, such that

$$D_{\text{KL}}(p \,\|\, p') \leq C\varepsilon^2.$$

**Proof.** *Step 1: Perturbation of the head output.* Masking removes $\Delta o = o - o' = \sum_{j \in S} a_{ij} v_j$. By the triangle inequality and boundedness of $v_j$,

$$\|\Delta o\| \leq \sum_{j \in S} a_{ij} \|v_j\| \leq C_v \sum_{j \in S} a_{ij} \leq C_v \varepsilon. \tag{1}$$

*Step 2: Propagation through the affine map.* Since $W$ is $L_W$-Lipschitz,

$$\|Wo - Wo'\| \leq L_W \|\Delta o\| \leq L_W C_v \varepsilon. \tag{2}$$

Let $\Delta z = Wo - Wo'$ denote the logit perturbation.

*Step 3: Softmax perturbation bound.* By local Lipschitz continuity of softmax,

$$\|p - p'\|_1 \leq L_{\text{sm}} \|\Delta z\| \leq L_{\text{sm}} L_W C_v \varepsilon. \tag{3}$$

*Step 4: KL divergence bound.* For distributions with $\min_y p'(y) \geq \delta$,

$$D_{\text{KL}}(p \,\|\, p') \leq \frac{1}{2\delta} \|p - p'\|_1^2. \tag{4}$$

Combining (3) and (4),

$$D_{\text{KL}}(p \,\|\, p') \leq \frac{(L_{\text{sm}} L_W C_v)^2}{2\delta} \varepsilon^2 =: C\varepsilon^2.$$

$\square$

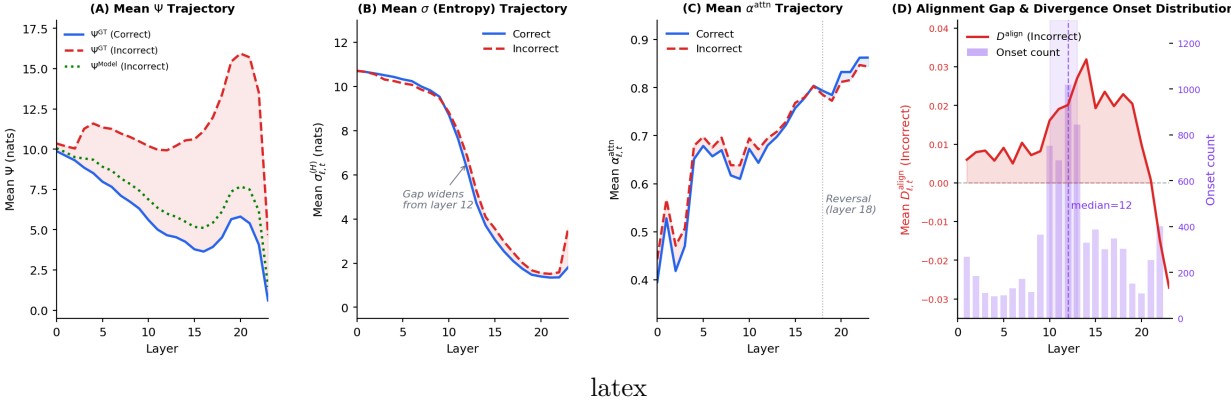

Figure 9: **Layer-wise trajectory of CDD proxies for Pythia-1.4B (200 C4 sequences; 16,211 correct and 16,211 incorrect positions).** **(A)** Mean $\Psi_{\ell,t}^{\mathrm{GT}}$ trajectory. Correct positions decrease broadly across layers; incorrect positions increase to a peak near layer 20 before converging at the final layer. The trajectory separation replicates the qualitative pattern of GPT-2 Large at reduced magnitude, consistent with shorter model depth. **(B)** Mean $\sigma_{\ell,t}^{(H)}$ (entropy) trajectory. Both conditions decrease broadly across layers. Unlike GPT-2 Large, the gap between conditions emerges only from layer 12 onward. **(C)** Mean $\alpha_{\ell,t}^{\mathrm{attn}}$ trajectory. Incorrect positions exhibit slightly higher concentration in layers 0–17, consistent with the excessive coherence regime. This relationship reverses in layers 18–23 (dotted line), suggesting architecture-dependent behavior in the deeper layers. **(D)** Mean residual alignment gap $D_{\ell,t}^{\mathrm{align}}$ (red line, left axis) and divergence onset distribution (bars, right axis) for incorrect positions only. The gap remains positive across layers 1–21; the onset distribution is predominantly unimodal, concentrated at layers 10–13 (median $= 12$), contrasting with the bimodal pattern observed in GPT-2 Large.

**Interpretation.** This bound shows that suppressing interactions whose total attention weight is $O(\varepsilon)$ induces only $O(\varepsilon)$ changes in logits and therefore only $O(\varepsilon^2)$ changes in KL divergence. Low-influence interactions are thus *dynamically negligible* in the sense of Proposition 1. The bound provides a conservative sufficient condition for stability: empirically, as shown in Experiment III (Section 5.4), semantic invariance persists far beyond the perturbative regime, even when the removed attention mass is no longer $O(\varepsilon)$.

## A.2 Cross-Architecture Replication: Pythia-1.4B

We replicate the core analysis on Pythia-1.4B (Biderman et al., 2023), a decoder-only model with 24 transformer blocks and approximately 1.4 billion parameters. The same experimental protocol is applied: teacher-forcing on 200 C4 English validation sequences, matched correct/incorrect positions (16,211 each), and identical proxy definitions as in Section 5.1. The replication is organized by role in the argument. Sections A.2.1–A.2.3 examine the empirical phenomena that C3 is required to unify (C1 and C2). Section A.2.4 extends the replication to the regime asymmetry of Experiment II (C3 Signal 2). Section A.2.5 replicates Experiment III on Pythia-1.4B, producing C3 Signal 3 and enabling direct assessment of three-way convergence in this architecture. Section A.2.6 reports the validation of the logit lens readout path for Pythia-1.4B.

### A.2.1 Layer-wise Trajectory of $\Psi^{\mathrm{GT}}$: Replication of $\Psi^{\mathrm{GT}}$ Divergence

The qualitative pattern observed in GPT-2 Large replicates in Pythia-1.4B: correct positions exhibit a broadly decreasing $\Psi^{\mathrm{GT}}$ trajectory while incorrect positions exhibit an increasing one, with sharp convergence at the final layer (Figure 9A). The absolute separation magnitude is smaller, consistent with Pythia-1.4B having fewer blocks over which divergence can accumulate, but the separation profile expressed as a fraction of model depth is qualitatively similar. $\Psi^{\mathrm{GT}}$ exceeds $\Psi^{\mathrm{Model}}$ at every layer for incorrect positions, confirming that the model assigns lower surprisal to its own incorrect prediction throughout inference. This trajectory separation constitutes the observational basis from which C3 Signal 1 (divergence onset) is derived, and its replication confirms that the underlying phenomenon is not specific to GPT-2 Large.

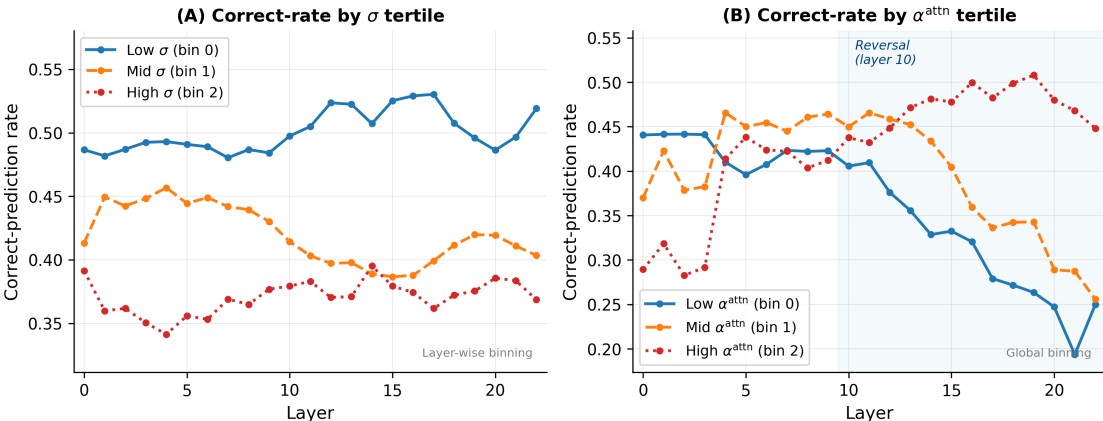

Figure 10: **Layer-wise correct-prediction rates stratified by tertile bins of (A) the diffusion proxy $\sigma_{\ell,t}^{(H)}$ and (B) the attention concentration proxy $\alpha_{\ell,t}^{\text{attn}}$ — Pythia-1.4B** (200 C4 validation sequences). Bin 0 denotes the lowest tertile, bin 2 the highest. **(A)** The lowest-$\sigma$ tertile exhibits consistently higher correct-prediction rates across all layers, with the gap widening from layer 12 onward, replicating the directional pattern observed in GPT-2 Large and consistent with C2. Bins are defined layer-wise to account for Pythia-1.4B's rapid entropy concentration in deeper layers, which renders global quantile boundaries uninformative. **(B)** The highest-$\alpha$ tertile is associated with lower correct-prediction rates in layers 0–9 and higher rates from layer 10 onward (shaded region), replicating the depth-dependent inversion formalized as C1. Bins are defined globally.

### A.2.2  Layer-wise Evolution of $\sigma$ and $\alpha^{\text{attn}}$: Replication of the Attention Paradox (C1)

$\sigma_{\ell,t}^{(H)}$ decreases broadly across layers in both conditions (Figure 9B). Unlike GPT-2 Large, the gap between correct and incorrect positions emerges only from layer 12 onward. The attention concentration proxy $\alpha^{\text{attn}}$ exhibits a higher value for incorrect positions in early-to-middle layers, but this relationship reverses in deeper layers (Figure 9C).

A complementary bin-level analysis corroborates these patterns (Figure 10). In Panel (A), the lowest-$\sigma$ tertile exhibits consistently higher correct-prediction rates across all layers, providing observational support for C2. In Panel (B), the highest-$\alpha$ tertile is associated with lower correct-prediction rates in layers 0–9 and higher rates from layer 10 onward, replicating the depth-dependent inversion of C1.

### A.2.3  Alignment Gap and Divergence Onset (C3 Signal 1 in Pythia-1.4B)

$D_{\ell,t}^{\text{align}}$ is positive for incorrect positions across the majority of layers (Figure 9D), consistent with GPT-2 Large. The onset detection rate is substantially lower (45.9% vs. 75.2%), consistent with the smaller alignment gap magnitude in this model. The onset distribution is predominantly unimodal, concentrated at layers 10–13, contrasting with the bimodal structure in GPT-2 Large. Expressed in relative depth, the median onset occurs at approximately 52% vs. 63% in GPT-2 Large—both near the middle of the model. The replication outcomes for Experiment I are summarized in Table 10.

### A.2.4  C2 Replication: Regime Asymmetry in Pythia-1.4B (C3 Signal 2)

This section examines whether the regime-dependent sign reversal of $\text{corr}(\sigma_{\ell,t}^{(H)}, \Delta\Psi_{\ell,t}^{\text{GT}})$, the empirical phenomenon formalized as C2 and contributing C3 Signal 2 in GPT-2 Large, replicates in Pythia-1.4B. The same post-hoc coupling analysis pipeline as Experiment II (Section 5.3) is applied, with `max_layer=22` to exclude the logit-lens boundary artifact. Table 11 summarizes the C2 replication results across both architectures.

The sign reversal replicates in Pythia-1.4B and the directional asymmetry between correct and incorrect positions is preserved across both architectures, confirming that C2, and the phenomenon it formalizes as a

Table 10: Replication of CDD qualitative predictions in Pythia-1.4B relative to GPT-2 Large (Experiment I), organized by role in the argument.

| Role | Structural Requirement | GPT-2 Large | Pythia-1.4B | Assessment |
|---|---|---|---|---|
| *C3 Signal 1 (primary contribution of Experiment I)* | | | | |
| C3 | $\Psi^{\mathrm{GT}}$ decreases with depth (correct) | Consistent | Consistent | Consistent |
| C3 | $\Psi^{\mathrm{GT}}$ increases with depth (incorrect) | Consistent | Consistent | Consistent |
| C3 | $D^{\mathrm{align}}$ positive for incorrect positions | Layers 1–34 | Layers 1–21 | Consistent |
| *C1: attention paradox (phenomenon unified by C3)* | | | | |
| C1 | $\alpha^{\mathrm{attn}}$–accuracy relationship inverts with depth | Layer 11 | Layer 10 | Consistent (qualitative) |
| *Supporting evidence (C2, C4)* | | | | |
| C2, C4 | $\sigma$ decreases with depth | Consistent | Consistent | Consistent |
| C2, C4 | $\sigma$ higher for incorrect positions | All layers | From layer 12 | Partially consistent |

Table 11: C2 replication: $\mathrm{corr}(\sigma, \Delta\Psi^{\mathrm{GT}})$ sign reversal between GPT-2 Large and Pythia-1.4B (C3 Signal 2).

| | GPT-2 Large | Pythia-1.4B |
|---|---|---|
| Total layers (analyzed) | 34 | 22 |
| Sign reversal (incorrect) | Layer 21 | Layer 19 |
| Relative depth of reversal | 60% | 83% |
| Duration of positive regime | 12 layers | 3 layers |
| Correct: sustained sign reversal | No | No |
| Directional asymmetry replicated | Yes | Yes |

contribution to C3 Signal 2, is not an artifact of a single model. The later relative depth of reversal (83% vs. 60%) and shorter positive regime (3 vs. 12 layers) may reflect architectural factors or the compressed post-transition depth in the shallower model; whether these quantitative differences reflect systematic architecture-dependent properties of regime transitions or are partially confounded by differences in depth and training data remains an open question for future investigation.

### A.2.5 Experiment III Replication: Dynamic Sparsity in Pythia-1.4B (C3 Signal 3)

This section replicates Experiment III on Pythia-1.4B using the same tail-mass pruning protocol as Section 5.4, with `keep_min=1` ensuring that at least one attention entry is retained per query position after pruning. Part A evaluates single-layer robustness at the final transformer block (layer 23) across pruning thresholds $\varepsilon \in \{0, 0.01, 0.02, 0.05, 0.1, 0.2, 0.3, 0.5\}$. Part B evaluates layer-wise robustness at all 24 transformer blocks with fixed $\varepsilon = 0.1$, producing C3 Signal 3 for Pythia-1.4B.

**Part A: Single-layer robustness at the final layer.** Table 12 reports the results at the final transformer layer. Across all tested thresholds, the output distribution remains highly stable. At $\varepsilon = 0.1$, approximately 99.3% of attention entries and 4.5% of attention mass are removed, yet KL divergence is $2.3 \times 10^{-5}$ and Top-1 agreement is 0.998. Even at $\varepsilon = 0.5$, where 99.8% of entries and 27% of mass are suppressed, Top-1 agreement remains at 0.99. The pattern is qualitatively identical to GPT-2 Large (Table 6), indicating that the sparsity structure of the final layer generalizes across architectures, and is consistent with Proposition 1.

Table 12: Part A: Threshold-invariant robustness at the final transformer layer (layer 23) of Pythia-1.4B under tail-mass pruning across all query positions.

| $\varepsilon$ | KL (mean) | Top-1 | Removed mass | Removed frac |
|---|---|---|---|---|
| 0.00 | 0.000 | 1.000 | 0.000 | 0.000 |
| 0.01 | $3.7 \times 10^{-7}$ | 1.000 | 0.007 | 0.968 |
| 0.02 | $1.3 \times 10^{-6}$ | 0.999 | 0.012 | 0.978 |
| 0.05 | $6.7 \times 10^{-6}$ | 0.999 | 0.026 | 0.988 |
| 0.10 | $2.3 \times 10^{-5}$ | 0.998 | 0.045 | 0.993 |
| 0.20 | $7.6 \times 10^{-5}$ | 0.996 | 0.076 | 0.996 |
| 0.30 | $1.5 \times 10^{-4}$ | 0.994 | 0.113 | 0.997 |
| 0.50 | $3.9 \times 10^{-4}$ | 0.990 | 0.270 | 0.998 |

Table 13: Part B: Layer-wise robustness under tail-mass pruning with $\varepsilon = 0.1$, `eval_mode=all`, `keep_min=1`, selected transformer layers of Pythia-1.4B. Full results across all 24 layers are shown in Figure 7B.

| Layer | KL (mean) | Top-1 | Removed mass | Removed frac |
|---|---|---|---|---|
| 0 | $1.61 \times 10^{-2}$ | 0.950 | 0.096 | 0.870 |
| 8 | $4.33 \times 10^{-4}$ | 0.988 | 0.077 | 0.967 |
| 16 | $1.71 \times 10^{-3}$ | 0.986 | 0.068 | 0.984 |
| **19** | $\mathbf{2.42 \times 10^{-3}}$ | **0.982** | **0.060** | **0.991** |
| 23 | $2.25 \times 10^{-5}$ | 0.998 | 0.045 | 0.993 |

**Part B: Layer-wise differential robustness (C3 Signal 3).** Table 13 and Figure 7B report KL divergence across all 24 transformer layers. Three patterns are noteworthy.

**Layer 0 is substantially more sensitive than all other layers.** At $\varepsilon = 0.1$, layer 0 exhibits KL $= 1.61 \times 10^{-2}$, approximately $713\times$ higher than the final layer ($2.25 \times 10^{-5}$), indicating that the attention distribution at the first transformer block is highly diffuse. This ratio is substantially larger than in GPT-2 Large ($9.3\times$) indicating that the steeper layer 0 sensitivity in Pythia-1.4B reflects architectural differences between the two models rather than a sequence-length artifact.

**KL exhibits a local peak at layer 19, constituting C3 Signal 3.** Among layers 1–23, KL reaches a local maximum at layer 19 ($2.42 \times 10^{-3}$), approximately $107\times$ higher than the final layer. This peak coincides with C3 Signal 2 (the $\sigma$–$\Delta\Psi$ sign reversal at layer 19; Appendix A.2.4): two substantively distinct measurement sources, logit-lens entropy and attention pruning robustness, identify the same critical depth. The coincidence of these two signals at a single layer provides stronger convergent evidence for a shared structural transition than the five-layer span observed in GPT-2 Large, even though C3 Signal 1 does not converge to the same depth.

**KL decreases monotonically from layer 19 to layer 23.** Following the peak at layer 19, KL decreases sharply: from $2.42 \times 10^{-3}$ at layer 19 to $8.31 \times 10^{-5}$ at layer 20 ($29\times$ drop in a single layer) and continues decreasing to the minimum at layer 23. This post-peak decrease mirrors the pattern observed in GPT-2 Large, where KL also decreases broadly following the pruning sensitivity peak.

Table 14: C3 signal depths in Pythia-1.4B compared to GPT-2 Large.

| Signal | GPT-2 Large | | Pythia-1.4B | |
|---|---|---|---|---|
| | Layer | Rel. depth | Layer | Rel. depth |
| C3 Signal 1 (divergence onset) | 22 | 63% | 12 | 52% |
| C3 Signal 2 ($\sigma$–$\Delta\Psi$ reversal) | 21 | 60% | 19 | 83% |
| C3 Signal 3 (pruning peak) | 17 | 49% | 19 | 83% |
| Convergence window | layers 17–22 (five-layer span) | | S2+S3 at layer 19; S1 at layer 12 | |

**Three-signal summary for Pythia-1.4B.** Table 14 summarizes the three C3 signal depths across both architectures. In Pythia-1.4B, C3 Signals 2 and 3 converge exactly at layer 19, while C3 Signal 1 occurs at layer 12, seven layers earlier (relative depth gap: 31%). This pattern differs from GPT-2 Large, where all three signals fall within a five-layer span. In Pythia-1.4B, incorrect trajectories begin to commit directionally at an earlier relative depth (layer 12), but the coupling and pruning signatures do not emerge until layer 19. The exact coincidence of Signals 2 and 3 at layer 19 satisfies the core evidential logic of C3: independently measured observables converge to a shared structural transition in this architecture as well. Whether the earlier position of Signal 1 reflects a genuine two-stage transition process or an architecture-specific feature of Pythia-1.4B's representational geometry is not resolved by the current data and remains a direction for future investigation.

### A.2.6 Logit Lens Readout Path Validation

The logit lens readout path was validated independently for each architecture against the model's actual output logits. For GPT-2 Large, the selected path is `lm_head`$(h_{\ell,t})$, applied directly to the intermediate hidden state without an additional layer normalization step; for Pythia-1.4B, the selected path is `embed_out`$(h_{\ell,t})$, applied analogously. In both cases, the selected path achieves exact agreement with the model's final-layer predictions across all evaluated samples, confirming the validity of the logit lens measurements reported in this appendix.

