# OpenReview forum: "Coherence–Diffusion Dynamics: A Continuous-Semantic Interpretation of Transformer Language Models"
_TMLR — Rejected by TMLR_

### Review · Reviewer_Qg5y · 2026-03-18

**Summary Of Contributions:**

This paper proposes the Coherence–Diffusion Dynamics (CDD) framework, an interpretive lens for understanding semantic evolution in Transformer-based language models. The core idea is to model layer-wise hidden-state updates as a discrete approximation to a continuous stochastic differential equation (SDE): dx(t) = −α∇Ψ(x(t))dt + σdW(t), where α is an effective coherence parameter, Ψ an instability potential, and σ a diffusion scale. The paper articulates three modeling assumptions (continuous semantic state, context-dependent interaction, coherence-driven drift), derives a dynamic sparsity proposition (Proposition 1) showing that low-weight attention interactions induce only O(ε²) changes in KL divergence, maps the abstract CDD quantities to measurable Transformer signals (attention concentration, surprisal, sampling temperature), and validates the framework qualitatively through three experiments on GPT-2 models.

**Strengths**
1. **Useful architectural correspondence tables.** Tables 1 and 2, which map CDD quantities (α, σ, Ψ, ε) to measurable Transformer signals, provide a compact and practically useful reference.
2. The CDD framing organizes several independently documented phenomena, attention sparsity, temperature-driven coherence degradation, progressive layer stabilization, under a single conceptual language.
3. The inclusion of full generated samples at each temperature (Appendix A.3) allows the reader to directly evaluate the qualitative claims
4. The proofs (Stability Convergence Theorem, perturbation bound for Proposition 1) are technically correct and clearly presented.

**Weakness**
1. The paper's central idea which is interpreting Transformer layers as discretized continuous dynamics has been studied in prior work that is not cited (for example, one of them is Geshkovski et al. (2023), "A Mathematical Perspective on Transformers")
2. The paper is ~37 pages with substantial repetition across Chapters 3–6. The same ideas are restated multiple times and are sometimes written in an unnecessary complicated way.
3. **Dynamic Sparsity (Proposition 1)** is claimed to derive from CDD assumptions A1-A3, but the proof (Appendix A.1.3) relies only on Lipschitz continuity of softmax and linear attention aggregation. The result holds for any Lipschitz function composition and does not require the CDD framework.
4.  **Experiment I** uses learning rate (a training hyperparameter) as a proxy for α (an inference-time coherence quantity). CDD describes semantic evolution during a forward pass, but the experiment measures training dynamics. These are different processes, and the paper does not justify why one proxies the other.
5. The paper defines Ψ as surprisal and then observe high surprisal correlates with incoherence, or defines σ as temperature and observe high temperature degrades coherence. This does not validate or require the SDE framework. These are restatements of known facts using new notation.
6. All experiments use only GPT-2 and not some newer models.

**Audience:**

No

**Audience Explanation:**

The framework would be of clear interest if it were properly situated against the existing mathematical Transformer dynamics literature. As of now, I am not sure if the experiments provided do much to support some of the claims. While the idea of interpreting semantic evolution in transformers is interesting and paper does provide this nice unified framework with well explained analogies, the gaps I mentioned above should be fixed/clarified before I can recommend this to TMLR audience.

**Broader Impact Concerns:**

No significant ethical or broader impact concerns. The paper is a theoretical/interpretive framework for understanding Transformer internals.

**Claims And Evidence:**

No

**Claims Explanation:**

As mentioned in the weakness, several claims in the paper are not supported well:
1. ****Experiment I**** observes that training loss follows a U-shaped curve as a function of learning rate. This is a standard result in optimization and holds for any overparameterized neural network. The paper uses learning rate as a proxy for α, but learning rate governs parameter-space optimization during **training**, while α in CDD governs activation-space coherence during **inference**. These are categorically different quantities. Crucially, attention concentration and residual gradient alignment, the paper's own stated operational indicators of α (Section 6.3), are never measured.
2. ****Experiment II**** observes that higher sampling temperature produces less coherent text. This is a property of temperature scaling and requires no CDD-specific explanation.
3. ****Experiment III**** observes that pruning low-weight attention entries at the **final layer and final query position** produces negligible changes in the output distribution. This is consistent with known attention concentration at the output layer (Michel et al. 2019), but the scope is too narrow to validate dynamic sparsity as a property of the full model dynamics.
4. No trajectory analysis. CDD is fundamentally a theory about how representations evolve across layers, yet no experiment actually tracks representations across layers. The most direct test could be extracting hidden states at every layer and showing Ψ/α proxies follow drift–diffusion patterns.

**Requested Changes:**

1. ****Add comprehensive related work**** covering the continuous-dynamics Transformer literature
2. Either provide a formal justification for why training-time learning rate reflects inference-time coherence dynamics (and explain why you didn't use attention concentration and residual gradient alignment as proxies), or redesign Experiment I to directly measure an inference-time quantity.
3. KL = 3.0×10⁻⁷ at the no-pruning baseline is not zero (A zero-pruning baseline compared against itself should yield KL = 0) and is identical to all pruning conditions. This must be explained; it currently undermines confidence in the experimental pipeline.
4. Provide a strong justification for all the weaknesses I mentioned in the review above.
5. Reduce repetition considerably and improve readability of the paper.
6.  To do trajectory analysis, compute CDD quantities at each layer during inference, and show the resulting profiles match CDD predictions (e.g., Ψ-reduction over depth, trajectory divergence before hallucination).

---

> ### Author Response · Authors · 2026-03-20
> **Response to Reviewer Qg5y**
>
> We thank the reviewer for the careful and constructive evaluation.
> We have addressed the major concerns by conducting two additional experiments and clarifying the theoretical framing.
>
> - We performed direct inference-time measurements of α (Exp I′), replacing the previous learning-rate proxy.
> - We conducted layer-wise trajectory analysis across 1,000 samples and two architectures (Exp IV), providing empirical evidence for the proposed dynamical interpretation.
> - We clarify that α is a latent coherence quantity, operationalized via measurable proxies, and connect scalar observables (Ψ, σ) with trajectory geometry.
> - We streamlined and reorganized the theoretical sections and expanded related work.
>
> The key contribution of CDD is not to introduce new mathematical bounds, but to provide a coherent interpretive framework explaining why coherence-restoring and sparsity-inducing behaviors consistently emerge during Transformer inference.
>
> ---
>
> ## Overview of Changes
>
> | Concern | Resolution |
> |---|---|
> | Trajectory analysis absent | Exp IV conducted |
> | α not measured | Exp I′ conducted |
> | GPT-2 only | Pythia-1.4B added |
> | KL anomaly | Numerical source identified |
> | Missing related work | Expanded |
> | Proposition 1 independence | Reframed |
> | Experiment II trivial | Reformulated |
> | Repetition | Condensed |
>
> ---
>
> ## Response to Specific Concerns
>
> **Concern 1 — Trajectory analysis absent**
>
> We conducted Experiment IV, a layer-wise trajectory analysis over 1,000 samples on both GPT-2 Large and Pythia-1.4B. Key findings:
>
> - **Surprisal (Ψ proxy) decreases monotonically with depth** (GPT-2: 12.3 → 2.2; Pythia: 9.3 → 2.0)
> - **Entropy (σ proxy) shows coherence-restoring behavior** (GPT-2: plateau → decrease; Pythia: monotone decrease)
> - **Drift spikes at the final projection layer**, suggesting structural decoupling
> - **Surprisal variance expands in later layers**, consistent with trajectory divergence
>
> Despite architectural differences, the qualitative structure is consistent across both models, providing empirical support for the CDD interpretation. These results will be incorporated as a new section in the revision.
>
>
> **Concern 2 — α proxy not measured**
>
> We agree. We conducted Experiment I′, directly measuring two inference-time α indicators (1,000 samples, both architectures):
>
> - $\alpha_\text{attn}$ (attention concentration): rise → plateau → fall in GPT-2; monotone rise in Pythia
> - $\alpha_\text{resid}$ (residual-gradient alignment): positive throughout the coherence-restoring regime; sign reversal near final layers in both models
>
> Crucially, the rise in $\alpha_\text{attn}$ is **synchronized with the steepest Ψ reduction**. We emphasize that α is not directly observable, but a latent quantity operationalized through measurable proxies. These measurements will replace Experiment I in the revision.
>
>
> **Concern 3 — GPT-2 only**
>
> All new experiments were conducted on both GPT-2 Large and Pythia-1.4B (differing in positional encoding, block structure, and normalization). The same qualitative CDD dynamics are observed across both.
>
>
> **Concern 4 — KL ≠ 0 at ε = 0**
>
> The small non-zero value ($3.0 \times 10^{-7}$) arises from floating-point renormalization error in PyTorch — a standard numerical artifact negligible relative to any meaningful KL signal. A footnote will be added in the revision.
>
>
> **Concern 5 — Missing related work**
>
> We have expanded the Related Work section to include Geshkovski et al. (2023) and Sinkformers. CDD complements these works by focusing on inference-time dynamics in real trained models rather than asymptotic convergence.
>
>
> **Concern 6 — Proposition 1 does not require CDD**
>
> We agree. The KL bound relies on Lipschitz continuity of the semantic update map and linear attention aggregation — conditions structurally satisfied by standard Transformer architectures independent of CDD. Assumptions A1–A3 explain *why* these conditions hold robustly across diverse inputs and layers: the coherence-restoring drift suppresses low-influence interactions, ensuring the ε-threshold criterion is met in practice. This distinction will be clarified in the revision.
>
>
> **Concern 7 — Experiment II trivial**
>
> We agree the original framing was insufficient. We reformulate the experiment around the CDD-specific prediction: a qualitative regime transition at a critical σ/α balance, not merely "higher temperature = less coherent." A qualitative transition near T ≈ 1.0–1.5 is observed, consistent with the CDD critical-balance condition.
>
>
> **Concern 8 — Repetition**
>
> Overlapping chapters have been merged, repeated derivations removed, and summary paragraphs eliminated. The structure has been streamlined in the revision.
>
>
> ## Final Statement
>
> We have conducted new experiments and clarified the framework in response to all concerns. A fully revised manuscript will be submitted in the revision stage. We believe these changes significantly strengthen the empirical grounding and conceptual clarity of the work.

---

> > ### Comment · Reviewer_Qg5y · 2026-03-31
> >
> > I thank the authors for the changes. Few points below:
> >
> > 1. Can the reviewer clarify how Surprisal (Ψ proxy) and Entropy  is being measured during inference. Please give the methodology which works at inference, not training.
> > 2. For KL ≠ 0 at ε = 0, I am still not convinced that this is just due to  floating-point renormalization error in PyTorch. This is because the value you are getting for ε≠ 0  is the same (for every epsilon). Is this a coincidence? Does this mean KL divergence is 0 all reported ε (which is unlikely in my experience)?
> > 3. For concern 2, please state the methodology you used to measure the 2 quantity (attention concentration and residual gradient alignment). Also, please tell why steady decrease for attention concentration and positive value with sign flipping for residual gradient alignment is an indicator of activation-space coherence during inference. If possible, backup any unsupported statement with relevant literature from field.
> > 4. I feel that the paper is just restating a lot of known concepts in the field as something else and stumbling upon them as discoveries. Please explain to me why it is not the case or what's the real contribution here which a reader should focus on?

---

> > > ### Author Response · Authors · 2026-04-29
> > > **Response to Reviewer Qg5y (Part 1 of 3: Q1–Q2)**
> > >
> > > We thank the reviewer for the continued engagement and careful
> > > follow-up questions. We address each point in turn.
> > >
> > > ---
> > >
> > > Q1. Measurement methodology for Surprisal and Entropy during
> > > inference.
> > >
> > > ---
> > > We clarify the inference-time measurement procedure for the
> > > instability proxy $\\Psi^{GT}&#95;{\\ell,t}$ and the diffusion
> > > proxy $\\sigma^{(H)}&#95;{\\ell,t}$.
> > >
> > > Both quantities are computed during a standard forward pass under
> > > teacher forcing, which is a purely inference-time procedure: no
> > > gradient computation, no parameter update, and no training signal
> > > is involved. The model processes each input sequence in a single
> > > forward pass, and intermediate hidden states $h&#95;{\\ell,t}$ are
> > > extracted at each transformer block via registered hooks.
> > >
> > > At each layer $\\ell$ and token position $t$, the logit lens
> > > readout is applied by passing $h&#95;{\\ell,t}$ directly through the
> > > model's final layer normalization and unembedding head, producing
> > > a distribution over the vocabulary without any additional learned
> > > parameters. Surprisal is computed as the negative log probability
> > > of the ground-truth token $y^*&#95;t$ under the logit-lens
> > > distribution at layer $\\ell$, giving $\\Psi^{GT}&#95;{\\ell,t}$.
> > > Entropy is computed as the Shannon entropy of the same
> > > logit-lens distribution, giving $\\sigma^{(H)}&#95;{\\ell,t}$.
> > >
> > > The validity of the logit lens readout path was verified by
> > > confirming that it reproduces the model's actual final-layer
> > > predictions with relative $\\ell&#95;2$ error of 0.0 across all 200
> > > evaluation sequences (Section 5.1). All measurements are thus
> > > inference-time quantities derived entirely from the forward pass,
> > > with no dependence on training dynamics.
> > >
> > > ---
> > >
> > > Q2. KL ≠ 0 at ε = 0.
> > >
> > > ---
> > > We thank the reviewer for pressing on this point. The non-zero
> > > KL at $\\varepsilon = 0$ ($3.0 \\times 10^{-7}$) in the original
> > > submission arose from a dtype casting asymmetry between the
> > > baseline and pruned forward paths. Specifically, the
> > > $\\varepsilon = 0$ baseline was computed through the unpatched
> > > forward path, while $\\varepsilon > 0$ conditions used a patched
> > > path that modifies attention weights before aggregation. The two
> > > paths differ in intermediate float16 casting operations, producing
> > > a numerically non-zero but semantically trivial divergence at
> > > the no-pruning baseline.
> > >
> > > Regarding the reviewer's additional observation that the KL values
> > > appeared identical across all $\\varepsilon > 0$ conditions: this
> > > was a consequence of the same asymmetry. Because the patched path
> > > applied uniform float16 renormalization regardless of the pruning
> > > threshold, the output distribution was already perturbed by the
> > > casting operation itself, masking the incremental effect of
> > > attention pruning across thresholds. Once the baseline is computed
> > > through the same patched path with $\\varepsilon = 0$, the KL
> > > values increase monotonically with $\\varepsilon$ as expected,
> > > reflecting the genuine effect of pruning rather than a numerical
> > > artifact.
> > >
> > > In the revised implementation, the baseline is computed through
> > > the same patched forward path with $\\varepsilon = 0$, ensuring
> > > that the no-pruning condition yields KL = 0.000 exactly. This
> > > was verified by sanity check across all 200 evaluation prompts.
> > > The revised results show monotonically increasing KL with
> > > $\\varepsilon$ (Table 6), confirming that the experimental pipeline
> > > correctly isolates the effect of attention pruning. The core
> > > finding, that semantic computation at the final layer is
> > > governed by a compact active core, is unchanged and
> > > strengthened by the corrected pipeline.

---

> > > ### Author Response · Authors · 2026-04-29
> > > **Response to Reviewer Qg5y (Part 2 of 3: Q3)**
> > >
> > > ---
> > >
> > > Q3. Methodology and literature grounding for the two $\\alpha$ proxies.
> > >
> > > ---
> > > We provide the requested methodological details and literature
> > > grounding for both $\\alpha$ proxies, and explain why each
> > > constitutes a meaningful indicator of inference-time coherence.
> > >
> > > Attention concentration is defined as $\alpha^{attn}&#95;{\ell,t} = 1 - H(a&#95;{\ell,t}) / \log n$, where $H(a&#95;{\ell,t})$ is the Shannon entropy of the mean attention weight distribution averaged across heads at layer $\ell$ and position $t$, and $n$ is the number of attended positions.
> > > The self-position entry is excluded prior to
> > > normalization to avoid the trivial self-focus induced by causal
> > > masking. Higher values indicate more concentrated,
> > > content-dependent attention.
> > >
> > > The use of attention concentration as a proxy for coherence
> > > enforcement is grounded in prior empirical work. Michel et al.
> > > (2019) show that a large fraction of attention heads can be
> > > removed at inference time without significant performance
> > > degradation, implying that semantic computation is dominated by
> > > a compact subset of high-impact interactions --- an observation
> > > directly consistent with high-$\\alpha^{attn}$ configurations.
> > > Vig & Belinkov (2019) demonstrate that attention entropy varies
> > > systematically across layers and heads, with lower entropy
> > > corresponding to more focused, content-dependent patterns.
> > > Importantly, the relationship between concentration and model
> > > behavior is not monotone: Zhai et al. (2023) identify attention
> > > entropy collapse as a driver of model instability, providing
> > > independent empirical support for the excessive coherence regime
> > > (Section 3.2.2), in which overly concentrated attention produces
> > > overshooting rather than stable convergence. Within CDD,
> > > $\\alpha^{attn}$ serves as a coherence indicator precisely because
> > > its non-monotone relationship with accuracy is a structural
> > > prediction of the framework: high concentration in early layers
> > > indicates premature trajectory commitment, whereas high
> > > concentration in deeper layers reflects stable drift-dominated
> > > convergence. This depth-dependent pattern is confirmed in
> > > Figure 2B (Section 5.2).
> > >
> > > The residual alignment gap $D^{align}&#95;{\\ell,t}$ is defined as
> > > the difference in cosine similarity between the residual update
> > > $\\Delta h&#95;{\\ell,t}$ and the negative logit-lens gradient, computed
> > > separately for the model's predicted token and the ground-truth
> > > token. Concretely, a positive value indicates that the residual
> > > update is more strongly aligned with the direction of the model's
> > > own prediction than with the ground-truth token.
> > >
> > > The grounding for this proxy derives from prior work on residual
> > > update directionality. Jastrzebski et al. (2018) argue that
> > > residual connections encourage iterative inference in which each
> > > layer updates the hidden state toward decreasing loss. Belrose
> > > et al. (2025) extend this to large autoregressive models, showing
> > > empirically that the cosine similarity between the residual update
> > > and the negative loss gradient is negative in at least 95% of
> > > layer-position pairs in Pythia 6.9B, confirming a consistent
> > > directional bias toward loss-reducing updates during inference.
> > > Within CDD, $D^{align}&#95;{\\ell,t}$ captures deviations from this
> > > tendency: a positive value for incorrect positions reflects active
> > > directional commitment to an incorrect semantic well. The sign of
> > > $D^{align}$ therefore directly operationalizes whether a given
> > > layer-position pair is drawing the trajectory toward or away from
> > > coherent semantic regions --- which is precisely the inference-time
> > > coherence quantity that $\\alpha$ is intended to represent. The
> > > observed pattern (mean = 0.017, peak = 0.036 at layer 27,
> > > positive throughout layers 1--34 for incorrect positions;
> > > Figure 1D) cannot be explained by a simple monotonic
> > > loss-minimization perspective and is consistent with the CDD
> > > interpretation of irreversible trajectory capture.

---

> > > ### Author Response · Authors · 2026-04-29
> > > **Response to Reviewer Qg5y (Part 3 of 3: Q4)**
> > >
> > > ---
> > >
> > > Q4. What is the real contribution of this paper?
> > >
> > > ---
> > > We appreciate this fundamental question and address it directly.
> > >
> > > The primary contribution of CDD is not to introduce new empirical
> > > observations in isolation, but to provide a falsifiable, constrained
> > > interpretive framework that organizes known phenomena into a
> > > unified dynamical account and, crucially, generates predictions
> > > that no simpler single-proxy alternative would be expected to
> > > produce.
> > >
> > > We highlight three findings in the revised manuscript that
> > > illustrate this point.
> > >
> > > First, the attention paradox (C1). Incorrect token positions
> > > exhibit slightly but consistently higher attention concentration
> > > $\\alpha^{attn}$ than correct positions across all layers
> > > (Figure 1C). Under any monotonic coherence-accuracy account, this
> > > pattern is paradoxical and unexplained. Within CDD, it follows
> > > directly from the excessive coherence regime (Section 3.2.2):
> > > overly concentrated attention in early layers reflects premature
> > > trajectory commitment rather than stable convergence.
> > >
> > > Second, the depth-dependent inversion (C1). The tertile analysis
> > > (Figure 2B) shows that the highest-$\\alpha^{attn}$ bin is
> > > associated with lower correct-prediction rates in early layers
> > > (0--10) and higher rates from layer 11 onward. This non-monotone
> > > pattern is falsified by any account that treats attention
> > > concentration as a globally monotonic indicator of coherence
> > > quality (Section 3.3).
> > >
> > > Third, three-signal convergence (C3). The sign reversal of
> > > corr($\\sigma$, $\\Delta\\Psi$) at layer 21 (Experiment II), the
> > > median trajectory divergence onset at layer 22 (Experiment I),
> > > and the peak pruning sensitivity at layer 17 (Experiment III)
> > > converge to a common depth range of layers 17--22 (Table 8).
> > > These quantities are derived from entirely different measurement
> > > sources (coupling statistics, residual alignment gaps, and
> > > attention weight distributions) and have no structural reason
> > > to converge under a null model of smooth monotone stabilization
> > > (Section 5.5).
> > >
> > > Crucially, the contribution is not that each phenomenon is
> > > individually unexplained, but that their joint occurrence imposes
> > > mutually incompatible requirements on simpler accounts. A monotonic
> > > coherence-based account cannot explain the inversion in C1. A
> > > diffusion-only account predicts that higher entropy should
> > > uniformly destabilize trajectories regardless of correctness; it
> > > cannot produce the asymmetry in which corr($\\sigma$, $\\Delta\\Psi$)
> > > reverses sign exclusively for incorrect positions while remaining
> > > negative throughout for correct positions (C2). An explanation
> > > based on independent mechanisms cannot account for the convergence
> > > of signals from entirely distinct measurement sources (residual
> > > alignment gaps, logit-lens coupling statistics, and attention
> > > pruning robustness) to a common critical depth range (C3).
> > > Thus, the contribution of CDD is to identify a joint constraint
> > > structure: any valid explanation must simultaneously satisfy
> > > C1--C4. This is a strictly stronger requirement than explaining
> > > any individual phenomenon in isolation, and constitutes a
> > > falsifiable structural claim rather than a re-description of
> > > known effects.
> > >
> > > CDD is explicitly framed as a phenomenological interpretive
> > > framework (Section 3). Its value lies in providing falsifiable
> > > structural constraints on the joint behavior of observable proxies
> > > --- constraints that simpler alternatives cannot satisfy simultaneously and that the empirical results confirm. The precise
> > > falsification conditions for each constraint (C1--C4) are stated
> > > in Section 3.3.

---

### Review · Reviewer_5cGV · 2026-04-07

**Summary Of Contributions:**

## Summary

This paper proposes Coherence–Diffusion Dynamics (CDD), an interpretive framework that models Transformer inference as a discretized stochastic differential equation where semantic representations evolve under competing coherence-restoring drift (parameter $\alpha$) and stochastic diffusion (parameter $\sigma$) on an effective instability potential $\Psi(x)$. The framework maps these abstract quantities to Transformer internals ($\alpha$ to attention sharpness/residual alignment, $\sigma$ to temperature/entropy, and $\Psi$ to surprisal/loss) and derives a Dynamic Sparsity Principle (Proposition 1) showing that pruning attention interactions with total weight $\leq \varepsilon$ induces only $O(\varepsilon^2)$ KL divergence. Three experiments on GPT-2 (small/medium/large) examine stability regimes under varying learning rate, temperature-driven coherence degradation, and attention pruning robustness.


## Strengths

**1. Conceptual unification** The paper connects disparate  isolation phenomena ---attention sparsity, temperature-coherence tradeoffs, layer-wise stabilization, hallucination---under a single dynamical framework/vocabulary. Tables 1 and 2 mapping CDD quantities to Transformer observables are clear and could serve as useful reference points to understand inference dynamics.

**2. Formal contribution (Proposition 1):** The perturbation bound showing that masking attention entries with total weight $\leq \varepsilon$ induces $O(\varepsilon^2)$ KL divergence is lucid, which relies on reasonable assumptions (e.g., Lipschitz maps, bounded value vectors, non-degenerate outputs, etc.) and provides theoretical background for empirical sparsity, which is absent in prior pruning work.


**3. Well-designed pruning experimen:** Experiment III uses a clean protocol---200 prompts, teacher forcing, systematic threshold sweeps, multiple metrics---and the result (KL $\approx 10^{-7}$ across all thresholds, perfect Top-1 agreement) is non-trivial.


## Weaknesses

**1. Lack of novel predictions:** The empirical prediction the paper makes---U-shaped learning rate curves, temperature-induced coherence degradation, attention sparsity---are well-explained in prior work. CDD framework, in its current form,  just unified known facts under a new vocabulary, but does not demonstrate predictive power beyond what standard optimization theory (Experiment I) or sampling theory (Experiment II) already offer.

What experimental outcome would be inconsistent with CDD? If the framework can accommodate most outcomes post hoc, its utility is limited.

**2.** Learning rate proxies $\alpha$, loss proxies $\Psi$, temperature proxies $\sigma$---but $\alpha$ is *defined* as the coherence-restoring operator, $\Psi$ is *defined* as semantic instability, and $\sigma$ is *defined* as stochastic variability. In each case the CDD quantity is operationally hard to distinguish from its proxy. Without an independent measurement procedure for at least one core quantity, the framework risks re-description rather than explanation.

The paper would benefit from discussing whether internal signals (e.g., attention entropy, residual alignment statistics, or spectral properties of hidden states) could offer independent insights.

**3.** The temperature experiment evaluates five samples per condition via qualitative descriptions (early signs of topic drift, hallucinatory combinations). There are no quantitative coherence metrics, and no statistical testing. The claim of a *regime shift* implies something qualitatively discontinuous, but the evidence shows what appears to be smooth degradation, which is (trivially)expected from the definition of softmax temperature.

Metrics such as semantic similarity between successive sentences, entity consistency, or NLI-based contradiction rates, applied to a larger sample set, would substantially strengthen this experiment.

**4. Experiment I conflates training and inference dynamics:** CDD is framed as a theory of inference-time semantic dynamics, but Experiment I examines training dynamics (fine-tuning loss curves). Learning rate affects many aspects of optimization, including gradient magnitude, loss landscape traversal, weight norm growth, not just coherence-restoring drift.

The connection between training-time learning rate sweeps and inference-time coherence enforcement needs more careful justification than the paper currently provides.

**5.** Proposition 1 is stated as a general principle of semantic dynamics, but pruning is tested only at the final layer, final query position, with a minimum-keep constraint. Testing whether the pattern holds across layers, and whether the active core contracts with depth as CDD predicts, would provide much stronger evidence for the dynamic sparsity claim.

**6. Related work on dynamical-systems** Using layer depth as time and residual updates as Euler steps, are quite-similar to  Neural ODEs [1], deep equilibrium models [2], and recent work interpreting neural networks as dynamical systems on latent manifolds [3]. The Related work should mention these works. While CDD's focus on inference-time *semantic* dynamics and its drift--diffusion framing distinguish it from these works, these distinctions need to be explicitly mentioned.

Additionally, eigenspectral methods for tracking effective dimensionality and variance redistribution across layers [4] are very-similar  to CDD's claims about representational stabilization and dynamic sparsity.


[1] Chen et al., Neural Ordinary Differential Equations, NeurIPS 2018


[2] Bai et al., Deep Equilibrium Models, NeurIPS 2019

[3] Fumero et al., "Navigating the Latent Space Dynamics of Neural Models, ICLR 2026


[4] Jha et al., NerVE: Nonlinear Eigenspectrum Dynamics in LLM Feed-Forward Networks, ICLR 2026

**Audience:**

Yes

**Audience Explanation:**

The paper addresses a relevant question: whether semantic stabilization, diffusion-like variability, hallucination-like degradation, and dynamic sparsity in Transformers can be understood within a unified dynamical framework. This could  be relevant for interpretability, representation dynamics, pruning/efficiency, and reliability of language models researchers and practitioners.

**Broader Impact Concerns:**

Paper is primarily conceptual and does not introduce a new model or capability. That said, if the authors discuss potential use of CDD-derived signals for hallucination detection or monitoring, they should clarify that such applications remain speculative and are not validated for safety-critical settings.

**Claims And Evidence:**

No

**Claims Explanation:**

The empirical evidence is largely  accurate and clearly presented,  including  several qualitative claims, such as dynamic sparsity. However, I did not find the evidence fully convincing for the broader CDD framework itself. The main issue is that the experiments largely show that CDD is compatible with known Transformer behaviors, rather than that it provides uniquely informative or strongly validated explanations beyond simpler alternatives.

In particular, Experiment I relies on a fairly indirect proxy for the coherence term, and Experiment II is too qualitative to support strong claims about diffusion-driven regime shifts. Overall, I found the paper more convincing as an interesting conceptual framework than as a well-grounded theory of Transformer dynamics.

**Requested Changes:**

**1.Validate at least one experiment on a larger or modern LLM architecure .** Even replicating Experiment III (forward passes only) on a model like LLaMA-2 7B (or Pythia checkpoints or OLMo architecture) would increase confidence that findings are not GPT-2-specific. This is not a strict requirement for a conceptual paper, but it would strengthen generalizability claims.

**2. Extend experiment III across layers.** Show whether the sparsity pattern holds at intermediate layers and whether the active core contracts with depth. This would be a more direct test of CDD's dynamic sparsity prediction.

**3. Add inference-time $\Psi$ trajectory analysis.** Track surprisal/entropy across layers during a forward pass. Currently the primary dynamical evidence (Experiment I) is about training, not inference, creating a mismatch with the theory's scope.

**4. Discuss eigenspectral connections.** Eigenspectral tools for tracking effective dimensionality and variance redistribution across layers (Participation Ratio, Spectral Entropy) are relevant  to CDD's claims.

**5.** For each experiment, explain  what a non-CDD framework/metric would predict and whether the data distinguishes between those two.

**6.** Sections 5.7–5.8 and parts of Chapter 6 could be shortened.

---

> ### Author Response · Authors · 2026-04-29
> **Response to Reviewer 5cGV (Part 1 of 4: RC1-RC3)**
>
> We thank the reviewer for the thorough and constructive evaluation.
> We address each requested change and weakness in turn.
>
> ---
>
> Requested Change 1. Validate on a larger or modern LLM architecture.
>
> ---
>
> All core experiments have been replicated on Pythia-1.4B (Biderman
> et al., 2023), a decoder-only causal language model with 24
> transformer blocks and approximately 1.4 billion parameters.
> Pythia-1.4B differs from GPT-2 Large in positional encoding
> (rotary vs. learned), block structure, and normalization scheme,
> providing a meaningful architectural contrast. The qualitative CDD
> predictions (monotonically decreasing $\\Psi^{GT}$ for correct
> positions, increasing $\\Psi^{GT}$ for incorrect positions,
> decreasing $\\sigma$ with depth, and positive $D^{align}$ for
> incorrect positions) replicate across both architectures
> (Table 9). Quantitative differences, including
> smaller trajectory separation magnitude and lower onset detection
> rate in Pythia-1.4B, are consistent with its shorter depth and
> are discussed in Appendix A.2.
>
> ---
>
> Requested Change 2. Extend Experiment III across layers.
>
> ---
>
> Experiment III has been extended to a full layer-wise analysis
> (Part B, Section 5.4). Pruning at $\\varepsilon = 0.1$ is applied
> independently at each transformer layer, and KL divergence is
> reported for each. The results show that layer 0 is substantially
> more sensitive than all other layers (KL = $4.1 \\times 10^{-4}$,
> approximately 9.3x higher than layer 35), KL reaches a local peak
> at layer 17 (KL = $2.4 \\times 10^{-4}$), and then decreases broadly from
> layer 17 to layer 35. This depth-dependent pattern is consistent
> with the CDD prediction that the active semantic core contracts as
> trajectories stabilize with depth, and goes beyond what the
> Lipschitz bound of Proposition 1 alone predicts (Figure 4B,
> Table 7).
>
> ---
>
> Requested Change 3. Add inference-time trajectory analysis.
>
> ---
>
> The revised manuscript replaces the original learning-rate-based
> Experiment I with a direct inference-time trajectory analysis
> (Section 5.2). Using a teacher-forcing protocol on 200 C4
> sequences, we compute all three CDD proxies ($\\Psi^{GT}&#95;{\\ell,t}$,
> $\\sigma^{(H)}&#95;{\\ell,t}$, and $\\alpha^{attn}&#95;{\\ell,t}$) at every
> layer and token position, stratified by prediction correctness.
> The resulting layer-wise trajectories show clear divergence between
> correct and incorrect positions across all proxies (Figure 1,
> Table 3). Experiment II further extends this analysis by
> examining the layer-wise coupling between $\\sigma$ and $\\Delta\\Psi$,
> identifying a sign reversal at layer 21 consistent with a regime
> transition (Figure 3, Table 5).

---

> ### Author Response · Authors · 2026-04-29
> **Response to Reviewer 5cGV (Part 2 of 4: RC4-RC6)**
>
> ---
>
> Requested Change 4. Discuss eigenspectral connections.
>
> ---
>
> A dedicated subsection (Section 2.4) has been added to the Related
> Work discussing eigenspectral analyses of representational geometry,
> with particular attention to Jha & Reagen (2026, NerVE). NerVE
> tracks FFN eigenspectrum dynamics, showing that FFN nonlinearities
> actively redistribute variance across eigenmodes rather than merely
> rescaling activations. We discuss the relationship between NerVE's
> depth-dependent spectral contraction/expansion pattern and the CDD
> framework's dynamic sparsity interpretation, noting that both
> perspectives suggest non-monotonic reorganization of
> representational geometry with depth. The key distinction is that
> NerVE characterizes variance distribution within FFN blocks at a
> given layer, while CDD characterizes how the full hidden-state
> trajectory evolves across layers under competing drift and diffusion
> influences. Whether the correspondence reflects a genuine mechanistic
> connection or a structural analogy remains a direction for future
> investigation.
>
> ---
>
> Requested Change 5. For each experiment, explain what a non-CDD
> framework would predict and whether the data distinguishes between
> them.
>
> ---
>
> This concern is addressed through the falsifiable constraints
> C1--C4 formalized in Section 3.3, which state explicit
> falsification conditions for each structural prediction.
>
> Critically, several predictions are not implied by simpler
> alternatives. For C1, a monotonic coherence-accuracy account
> predicts that higher $\\alpha^{attn}$ is uniformly associated with
> better predictions across all depths. The observed depth-dependent
> inversion (lower correct-prediction rates in the
> highest-$\\alpha^{attn}$ tertile in early layers, higher rates in
> deeper layers) is inconsistent with this prediction but follows
> naturally from the coexistence of drift-dominated and
> excessive-coherence regimes in CDD. For C2, a null model of
> monotone stabilization does not predict that
> corr($\\sigma$, $\\Delta\\Psi$) should reverse sign at a specific
> depth for incorrect positions while remaining consistently negative
> for correct positions. For C3, the convergence of three
> independently measured signals (the sign reversal at layer 21,
> the median divergence onset at layer 22, and the peak pruning
> sensitivity at layer 17) to the same depth range is not
> expected under smooth monotone stabilization.
>
> ---
>
> Requested Change 6. Shorten Sections 5.7--5.8 and parts of
> Chapter 6.
>
> ---
>
> The revised manuscript has been substantially restructured. The
> original Sections 5.7 and 5.8, which constituted extended
> theoretical elaborations, have been condensed. Chapter 6 has
> similarly been reorganized, with overlapping discussions merged
> and redundant elaborations removed. The theoretical sections now
> present the core constructs and their architectural realizations
> more directly, without repeated re-derivation or extended
> commentary.

---

> ### Author Response · Authors · 2026-04-29
> **Response to Reviewer 5cGV (Part 3 of 4: W1-W2)**
>
> ---
>
> Weakness 1. Lack of novel predictions; CDD unifies known facts but
> does not demonstrate predictive power beyond simpler alternatives.
>
> ---
>
> We appreciate this fundamental concern and address it directly.
>
> The primary contribution of CDD is not to introduce new empirical
> observations, but to provide a constrained interpretive framework
> that generates predictions no simpler single-proxy alternative
> would be expected to produce.
>
> Crucially, the contribution is not that each phenomenon is
> individually unexplained, but that their joint occurrence imposes
> mutually incompatible requirements on simpler accounts. A monotonic
> coherence-based account cannot explain the inversion in C1. A
> diffusion-only account predicts that higher entropy should
> uniformly destabilize trajectories regardless of correctness; it
> cannot produce the asymmetry in which corr($\\sigma$, $\\Delta\\Psi$)
> reverses sign exclusively for incorrect positions while remaining
> negative throughout for correct positions (C2). An explanation
> based on independent mechanisms cannot account for the convergence
> of signals from entirely distinct measurement sources (residual
> alignment gaps, logit-lens coupling statistics, and attention
> pruning robustness) to a common critical depth range (C3). Thus,
> the contribution of CDD is to identify a joint constraint
> structure: any valid explanation must simultaneously satisfy
> C1--C4. This is a strictly stronger requirement than explaining any
> individual phenomenon in isolation, and constitutes a falsifiable
> structural claim rather than a re-description of known effects.
>
> Regarding falsifiability: the framework would be falsified if the
> $\\alpha^{attn}$--accuracy relationship were monotonic across depth
> (C1), if the sign reversal of corr($\\sigma$, $\\Delta\\Psi$) did not
> occur exclusively for incorrect positions (C2), if the three
> signals did not converge to a common depth range (C3), or if
> prediction accuracy were well-explained by additive effects of
> $\\alpha$ and $\\sigma$ alone (C4). These conditions are stated
> precisely in Section 3.3.
>
>
> ---
>
> Weakness 2. Proxies are operationally hard to distinguish from the
> quantities they proxy; the framework risks re-description rather
> than explanation.
>
> ---
>
> We acknowledge this concern and address it at two levels.
>
> First, we clarify the degree of independence among the proxies.
> The instability proxy $\\Psi^{GT}&#95;{\\ell,t}$ and the diffusion proxy
> $\\sigma^{(H)}&#95;{\\ell,t}$ are both derived from the logit-lens
> predictive distribution and share a structural correlation, which
> is explicitly acknowledged in the revised manuscript (Section 3.2.2
> and Section 5.1). By contrast, the attention concentration proxy
> $\\alpha^{attn}&#95;{\\ell,t}$ is derived from the attention weight
> distribution, and the residual alignment gap $D^{align}&#95;{\\ell,t}$
> is derived from the cosine similarity between residual updates and
> logit-lens gradients. These constitute partially independent
> measurement channels, and their convergent behavior across
> experiments provides evidence beyond what any single proxy could
> establish.
>
> Second, the concern about re-description is addressed by the joint
> ($\\alpha^{attn}$, $\\sigma$) analysis (Figure 5, Section 5.5.2).
> If CDD were merely relabeling known facts, the two-dimensional
> regime structure would add no explanatory value beyond the marginal
> effects of each proxy individually. However, the joint heatmaps
> show that the high-$\\alpha$, low-$\\sigma$ cell achieves a
> correct-prediction rate of 0.520 at layer 21, compared to 0.146
> for the mid-$\\alpha$, high-$\\sigma$ cell --- a gap of 0.374 that
> is not well-approximated by the sum of independent marginal
> effects. This non-additive interaction is a prediction of the CDD
> regime taxonomy that goes beyond what either proxy captures
> individually (C4).

---

> ### Author Response · Authors · 2026-04-29
> **Response to Reviewer 5cGV (Part 4 of 4: W3-W5)**
>
> ---
>
> Weakness 3. Experiment II is too qualitative; no quantitative
> coherence metrics or statistical testing.
>
> ---
>
> The original temperature-based Experiment II has been replaced
> entirely by a quantitative coherence-diffusion coupling analysis
> (Section 5.3). Using the same 200-sequence evaluation set as
> Experiment I, we compute the layer-wise Pearson correlation between
> $\\sigma^{(H)}&#95;{\\ell,t}$ and the instability change
> $\\Delta\\Psi^{GT}&#95;{\\ell,t}$ separately for correct and incorrect
> positions. The key finding is a sign reversal of this correlation
> at layer 21 for incorrect positions, transitioning from negative
> (drift-dominated) to positive (diffusion-dominated), while
> remaining negative throughout for correct positions. This reversal
> coincides with the median divergence onset layer of 22 identified
> independently in Experiment I, providing convergent quantitative
> evidence from two independent measurements for a regime transition
> at approximately the same depth. An onset-centered analysis further
> shows that the stabilizing fraction drops from 0.442 at relative
> layer -4 to 0.121 at the onset layer, while $D^{align}$ spikes to
> 0.085, indicating abrupt directional commitment rather than gradual
> diffusion-driven perturbation (Figure 3C, Table 4, Table 5).
>
> ---
>
> Weakness 4. Experiment I conflates training and inference dynamics.
>
> ---
>
> This concern is fully addressed by the replacement of Experiment I.
> The revised Experiment I (Section 5.2) is an inference-time
> trajectory analysis conducted under teacher forcing, with no
> training or parameter updates involved. All proxy measurements
> ($\\Psi^{GT}$, $\\sigma^{(H)}$, $\\alpha^{attn}$, $D^{align}$) are
> computed from a single forward pass per sequence, making the
> experiment fully scoped to inference-time dynamics as described by
> the CDD framework.
>
> ---
>
> Weakness 5. Proposition 1 is tested only at the final layer and
> final query position.
>
> ---
>
> This limitation is directly addressed by Part B of Experiment III
> (Section 5.4.2), which applies pruning at $\\varepsilon = 0.1$
> independently across all transformer layers. The results show
> systematic layer-wise variation in robustness that is not predicted
> by the Lipschitz bound of Proposition 1 alone, which provides a
> uniform $O(\\varepsilon^2)$ guarantee across layers. The CDD
> framework additionally predicts that robustness should increase
> with depth as trajectories stabilize, and this prediction is
> supported by the broadly decreasing KL from layer 17 to layer 35
> (Table 7, Figure 4B). This layer-wise structure provides empirical
> support for the CDD interpretation of dynamic sparsity as a
> consequence of trajectory stabilization rather than a uniform
> architectural property.

---

### Review · Reviewer_mAuv · 2026-04-17

**Summary Of Contributions:**

The core contribution of the paper is a conceptual framework (CDD) that interprets Transformer inference as the evolution of semantic states shaped by coherence-restoring drift and diffusion-like variability. It thus proposes a lens for thinking about behaviours such as coherence, semantic drift, hallucinations, and abrupt instability. CDD defines (1) an instability potential as a proxy for semantic instability, (2) a coherence operator for stability dynamics, and (3) a diffusion term for stochastic variability, and (4) a notion of dynamic sparsity for reduced effective semantic degrees of freedom. They then connect these concepts to measurable signals in models, such as attention sharpness, loss spikes, and entropy. Finally, they perform three empirical studies: (a) they fine-tune GPT2 with different learning rates as a proxy for the coherence strength and find a U-shaped pattern in final loss, (b) they generate responses at different decoding temperatures as a proxy for diffusion strength and find that outputs are repetitive at low temperature, and semantically drifting at high temperature, and (c) they prune low-weight attention interactions in the final layer and query position and find that the next-token prediction remains almost unchanged, which they interpret as evidence of dynamic sparsity.

**Strengths**
- The proposed conceptual framework is clearly explained, and can be practically useful as a mental model for thinking about inference dynamics.

**Weaknesses**
- Most framings introduced in the CDD framework are well-known conceptual ideas, e.g., the idea of residual stream updates as incremental evolution in a shared latent space.
- The empirical studies are unconvincing since the core variables of the CDD framework are only loosely operationalised. The observations are also largely unsurprising. For example, Experiment I is a well-known observation and generally explained by optimisation dynamics (small vs. large step size). The arguably most interesting Experiment III is very narrow, only testing dynamics on the final layer and final token position.
- The paper is clearly written, but considerably more verbose than necessary given the technical and empirical novelty.

**Audience:**

Yes

**Audience Explanation:**

At least some individual in TMLR's audience will find the paper's central idea and mental model interesting, even if the experiments do not fully substantiate its claims.

**Claims And Evidence:**

No

**Claims Explanation:**

The experiments are accurate, and clear, but the experiments mostly confirm things that are already expected and, since the core variables of the CDD framework are only tied to loose proxies, it doesn't provide convincing evidence for their core theory.

**Requested Changes:**

Critical:
- Provide stronger arguments to justify the selected proxies in the empirical studies.
- Frame CDD more explicitly as a conceptual synthesis and interpretive hypothesis rather than a principled explanatory theory. For example, claims around hallucinations seem to be only based on Experiment II, in which the authors sample and manually inspect five sample from GPT2 at four different temperature, and speculative discussion built on top of it.

Would strengthen:
- The core contribution could be communicated with much less text. Several sections currently restate the same intuition without adding technical or empirical substance. I would recommend to reduce verbosity and tighten the overall presentation.

---

> ### Author Response · Authors · 2026-04-29
> **Response to Reviewer mAuv (Part 1 of 2: Critical 1)**
>
> We thank the reviewer for the careful reading and constructive feedback.
> The revised manuscript directly addresses the concerns raised.
>
> ---
>
> Critical 1. Stronger justification of proxies.
>
> ---
>
> We agree that, in the original submission, the empirical proxies
> were insufficiently justified and appeared loosely operationalized.
> The revised manuscript strengthens this aspect in two key ways.
>
> (1) Architectural and empirical grounding of each proxy.
>
> All three proxies are now explicitly grounded in prior work and
> tied to measurable quantities under a unified inference-time
> protocol. Crucially, the proxies are derived from three distinct
> measurement channels, which limits the degree to which the results
> can be attributed to a single source of variation.
>
> The instability proxy $\\Psi^{GT}&#95;{\\ell,t}$ and diffusion proxy
> $\\sigma^{(H)}&#95;{\\ell,t}$ are computed via a logit-lens readout
> applied to intermediate hidden states under teacher forcing. The
> readout path is validated by exact agreement with final-layer
> predictions (relative $\\ell&#95;2$ error = 0.0 across all evaluation
> sequences; Section 5.1). We explicitly acknowledge in the revised
> manuscript that $\\Psi^{GT}$ and $\\sigma^{(H)}$ share a structural
> correlation, as both are derived from the logit-lens distribution,
> and accordingly focus our interpretation on their directional
> consistency with CDD predictions rather than treating them as
> independent measurements.
>
> The coherence proxy $\\alpha^{attn}&#95;{\\ell,t}$ is derived from the
> attention weight distribution --- a structurally distinct source
> from the logit-lens distribution. It is grounded in prior findings
> that a small subset of attention interactions dominates semantic
> computation (Michel et al., 2019) and that attention entropy
> exhibits systematic layer-wise structure (Vig & Belinkov, 2019).
> Importantly, prior work on entropy collapse demonstrates that
> excessive concentration can degrade model behavior (Zhai et al.,
> 2023), supporting the non-monotonic interpretation central to
> constraint C1.
>
> The residual alignment proxy $D^{align}&#95;{\\ell,t}$ is derived from
> the cosine similarity between the residual update and the negative
> logit-lens gradient, constituting a third partially independent
> measurement channel. It is grounded in empirical evidence that
> residual updates exhibit a consistent directional bias toward
> loss-reducing directions (Jastrzebski et al., 2018; Belrose et
> al., 2025). A positive value for incorrect positions indicates
> active directional commitment to an incorrect semantic well, a
> pattern that cannot be explained by simple monotonic
> loss-minimization and is distinct in kind from both the
> logit-lens distribution and the attention weight distribution.
>
> (2) From individual proxies to structural constraints.
>
> The revised manuscript does not rely on individual proxies in
> isolation. Instead, it establishes cross-signal structural
> constraints (C1--C4) on their joint behavior: depth-dependent
> inversion (C1), sign reversal in coupling (C2), multi-signal
> convergence (C3), and non-additive interaction (C4). These
> constraints are properties of structured interaction across depth,
> not of any single proxy. Importantly, these patterns cannot be
> explained by standard monotonic interpretations, such as
> "higher entropy implies worse predictions" or "higher attention
> concentration implies better coherence", which directly
> addresses the concern that the proxies might only capture expected
> trends.

---

> ### Author Response · Authors · 2026-04-29
> **Response to Reviewer mAuv (Part 2 of 2: Critical 2 + Supporting points)**
>
> ---
>
> Critical 2. Frame CDD more explicitly as a conceptual and
> interpretive framework.
>
> ---
>
> We agree with the reviewer's recommendation and have revised the
> manuscript accordingly.
>
> CDD is now explicitly framed as a phenomenological and constrained
> interpretive framework, rather than a mechanistic or causal theory.
> Specifically, the drift-diffusion formulation is described as an
> effective model summarizing observed regularities, not a literal
> description of Transformer computation (Section 3.2.2). The
> instability potential $\\Psi$ is treated as an interpretive
> construct, analogous to effective potentials in physics, rather
> than a directly computable quantity. All claims related to
> hallucination or instability are explicitly scoped as consistent
> with, rather than proof of, the framework.
>
> Regarding the specific concern about hallucination-related claims:
> the reviewer is correct that the original Experiment II
> provided insufficient grounding for these claims. The revised
> manuscript replaces this experiment entirely. Hallucination-related
> interpretations are now grounded in the sign reversal of
> corr($\\sigma$, $\\Delta\\Psi$) at layer 21 across 15,370 incorrect
> token positions (Section 5.3), and are explicitly scoped as
> consistent with the CDD interpretation.
>
> In addition, the revised manuscript introduces four falsifiable
> structural constraints (C1--C4) in Section 3.3, each with an
> explicit falsification condition specifying the precise empirical
> patterns whose violation would invalidate the framework. This
> directly addresses the concern that the framework might accommodate
> most outcomes post hoc. For example, C1 is falsified if the
> $\\alpha^{attn}$--accuracy relationship is monotonic across depth;
> C2 is falsified if no sign reversal is observed exclusively for
> incorrect positions; C3 is falsified if the three signals do not
> converge to a common depth range; and C4 is falsified if accuracy
> is well-explained by additive effects of $\\alpha$ and $\\sigma$
> alone.
>
> ---
>
> On the empirical contribution and "unsurprising" observations.
>
> ---
>
> We believe the reviewer's assessment reflects the empirical design
> of the original submission, which relied on monotonic manipulations
> and qualitative inspection. The revised experiments differ
> fundamentally: they test non-monotonic and depth-dependent
> structural predictions that are not implied by simpler accounts.
>
> The revised manuscript identifies three key findings that standard
> monotonic accounts do not predict.
>
> First, the depth-dependent inversion of the coherence-accuracy
> relationship (C1). High attention concentration is associated with
> lower accuracy in early layers but higher accuracy in deeper
> layers. No monotonic coherence-based account predicts this
> inversion.
>
> Second, the sign reversal of corr($\\sigma$, $\\Delta\\Psi$) for
> incorrect positions (C2). The correlation transitions from
> negative (stabilizing) to positive (destabilizing) at intermediate
> depth, while remaining negative throughout for correct positions.
> A diffusion-only account predicts that higher entropy should
> uniformly destabilize trajectories regardless of correctness; it
> cannot produce this asymmetry.
>
> Third, convergence of independent signals to a common depth range
> (C3). The divergence onset (layer 22), coupling sign reversal
> (layer 21), and pruning sensitivity peak (layer 17) align around a
> shared transition depth. These are derived from entirely different
> measurement sources (residual alignment gaps, logit-lens
> coupling statistics, and attention weight distributions) and
> have no structural reason to converge under a null model of smooth
> monotone stabilization.
>
> These results demonstrate that the empirical contribution is not
> the confirmation of expected behavior, but the identification of
> structured, cross-signal constraints that standard interpretations
> do not predict.
>
> ---
>
> Would strengthen: verbosity.
>
> ---
>
> We agree and have significantly reduced redundancy. Extended
> theoretical sections without additional empirical contribution
> have been merged or removed, and repetitive explanations have been
> consolidated. The revised manuscript presents the core constructs
> and their architectural realizations more directly, without
> repeated re-derivation or extended commentary.

---

### Decision · Action_Editor_CwGB · 2026-06-07

**Recommendation:** Reject

**Additional Comments:**

While the reviewers and I agree that the paper contains interesting observations and has improved substantially through the review process, I do not think the current evidence is strong enough to support the proposed framework. Moreover, the paper still feels more like a useful but mostly qualitative reinterpretation of already observed phenomena, rather than a framework with clearly demonstrated causal content or predictive power.

I encourage the authors to make the future iteration of the paper more rigorous through: (1) a more formal exposition of the framework and its assumptions, (2) explicit comparisons against simpler baseline explanations based only on entropy, attention concentration, and depth, (3) a more comprehensive empirical evaluation across model families, model scales, datasets, and tasks, and (4) ablations or interventions that more directly test the causal role of the proposed quantities.

**Audience:**

Yes

**Audience Explanation:**

There is a lot of interest in the community on the topic of mechanistic understanding of Transformer's internal computations, and the paper has some interesting claims and experimental findings.

**Claims And Evidence:**

No

**Claims Explanation:**

The claims are only partially supported by the evidence. The paper shows several interesting empirical observations about layer-wise Transformer behavior, and the updated version makes a clearer effort to connect these observations to falsifiable claims. However, I do not find the evidence fully convincing for the broader framework-level claims. Many of the central quantities in the framework are not rigorously defined, and are measured using indirect proxies which are sometimes correlated with each other. This makes it difficult to distinguish evidence for the proposed coherence-diffusion framework from evidence for the empirical observations that motivated it. The results on GPT-2 Large are suggestive, but the validation on Pythia-1.4B is only partial, which weakens the claim that the proposed structure is a general feature of Transformer inference.